

An Evaluation of the Performance of Sea-Bird Scientific's Autonomous SeaFET[TM]:
Considerations for the Broader Oceanographic Community
Cale A. Miller[1,3], Katie Pocock[2], Wiley Evans[2], and Amanda L. Kelley[1*]
1. College of Fisheries and Ocean Sciences, University of Alaska Fairbanks, Fairbanks, AK,
USA
2. Hakai Institute, Heriot Bay, BC, Canada
3. **Present address:** Department of Evolution and Ecology, College of Biological Sciences,
University of California Davis, CA, USA
*Correspondence to: Amanda L. Kelley** (alkelley@alaska.edu)
**Abstract**
The commercially available Sea-Bird SeaFET[TM] provides an accessible way for a broad
community of researchers to study ocean acidification and obtain robust measurements of
seawater pH via the use of an *in situ* autonomous sensor. There are pitfalls, however, that have
been detailed in previous best practices for sensor care, deployment, and data handling. Here, we
took advantage of two distinctly different coastal settings to evaluate the Sea-Bird SeaFET[TM] and
examine the multitude of scenarios in which problems may arise confounding the accuracy of
measured pH. High-resolution temporal measurements of pH were obtained during 3- to 5-month
field deployments in three separate locations (two in south-central, Alaska, USA, and one British
Columbia, Canada) spanning a broad range of nearshore temperature and salinity conditions.
Both the internal and external electrodes onboard the SeaFET[TM] were evaluated against robust
benchtop measurements for accuracy utilizing either the factory calibration, an *in situ* single-
point calibration, or *in situ* multi-point calibration. In addition, two sensors deployed in parallel
in Kasitsna Bay, AK, USA, were compared for inter-sensor variability in order to quantify other
factors contributing to SeaFET[TM] intrinsic inaccuracies. Based on our results, the multi-point
calibration method provided the highest accuracy (< 0.025 difference in pH) of pH when
compared against benchtop measurements. Spectral analysis of time series data showed that
during spring in Alaskan waters, a range of tidal frequencies dominated pH variability, while
seasonal oceanographic conditions were the dominant driver in Canadian waters. Further, it is
suggested that spectral analysis performed on initial deployments may be able to act as an *a
posteriori* method to better identify appropriate calibration regimes. Based on this evaluation, we
provide a comprehensive assessment of the potential sources of uncertainty associated with
accuracy and precision of the SeaFETs[TM] electrodes.
**1 Introduction**
The intrusion of excess anthropogenic $CO_2$ into the global oceans—referred to as ocean
acidification (OA)— induces a series of geochemical reactions that increases seawater $[H^+]$
(lowering pH) while concomitantly reducing the ocean's overall buffering capacity by reducing



the [$CO_3^{2-}$] (Caldeira and Wickett, 2003; Orr et al., 2005). Due to more dynamic natural physical
and chemical processes in the coastal ocean, a differentiation exists between open-ocean
acidification and nearshore coastal acidification. Open-ocean acidification of surface waters is
predominately a function of equilibration with atmospheric $p$CO$_2$, thus increasing on yearly and
decadal timescales as continued burning of fossil fuels ensues (Hofmann et al., 2011; Orr et al.,
2005). Coastal acidification, however, can manifest on short time and space scales driven by
riverine input and its chemical constituents (e.g., organic carbon, nutrients, and organic
alkalinity), community metabolism and organization, tidal cycles, upwelling, and groundwater
input (Duarte et al., 2013; Sunda and Cai, 2012; Waldbusser and Salisbury, 2014), all of which
can act in conjunction with increasing atmospheric CO$_2$, leading to more frequent, intense, and
longer-lasting acidification events (Hales et al., 2016; Harris et al., 2013). In the face of rapidly
changing coastal conditions, tracking and quantifying the progression of OA requires precise and
accurate measurements of carbonate chemistry over long periods of time; these can be achieved
by appropriately constraining the carbonate system by measuring at least two of the system's
parameters: total dissolved inorganic carbon (TCO$_2$), total alkalinity (TA), pH, and the partial
pressure of CO$_2$ ($p$CO$_2$). Despite the marked increase in OA research over the past decade
(Riebesell and Gattuso, 2015; Rudd, 2017), nearshore monitoring efforts—particularly in
estuarine waters—have been slow to ramp up, however, efforts are beginning to intensify as
technological advancements are made (Feely et al., 2010, 2016; Hales et al., 2016; Harris et al.,
2013; Newton et al., 2012; Waldbusser and Salisbury, 2014; Chan et al., 2017).
Acidification of Alaskan coastal waters is predicted to progress rapidly relative to other
regions within the next 50 years, and negatively impact the social-ecological structure of Alaskan
marine resources by disrupting the Alaska Native subsistence and commercial fisheries (Ekstrom
et al., 2015; Mathis et al., 2015b). The ocean waters present along the Alaskan coastline
experience chemical and physical drivers of seawater chemistry that are unique to this region.
The low seawater temperatures inherently have higher concentrations of dissolved CO$_2$, and
chemical and physical oceanic processes unique to Alaskan waters such as sea ice melt, glacial
discharge, and benthic pelagic coupling across shallow shelves are likely to exacerbate
acidification in this region (Evans et al., 2014; Mathis et al., 2011a, 2011b, 2012). Recently, an
OA monitoring initiative has been setup by the Alaska Ocean Observing Network (AOOS) to
track and provide accessible material dedicated to acidification research in Alaskan waters
(http://www.aoos.org/alaska-ocean-acidification-network). Along the Pacific coast of Alaska, a
robust benchtop system known as a Burke-o-Lator (BoL), which measures TCO$_2$ and $p$CO$_2$
either continuously in a flow-through environment or from discrete seawater samples (Bandstra
et al., 2006; Barton et al., 2012; Hales et al., 2016) has been installed in several locations,
including the OceansAlaska Shellfish Hatchery in Ketchikan, the Alutiiq Pride Shellfish
Hatchery in Seward (Evans et al., 2015), and at the Sitka Tribe of Alaska Environmental
Research Center (real-time data from Alaskan and other BoLs:
http://www.ipacoa.org/Explorer?action=oiw:fixed_platform). Nominal analytical uncertainty for
TCO$_2$ determinations from this system is 0.2% based on the reproducibility of sample and
certified reference material (CRM; provided by A. Dickson analyses). For $p$CO$_2$ determinations,
analytical uncertainty is 1.5% based on the inaccuracy of calculated CRM alkalinity relative to
the certified value. While the BoL has significant advantages for achieving robust OA
measurements in nearshore waters, the physical constraints of a benchtop system limit the spatial
dimension of which carbonate chemistry parameters can be measured. One potential resolution



to diminish the gap in coverage of OA monitoring is to utilize autonomous pH sensors, which are
far more versatile in their ability to monitor hard-to-reach areas.

Recent assessments regarding OA monitoring efforts have specifically highlighted the
benefits of accessibility by the commercially produced SeaFET[TM] pH sensor utilizing Honeywell
Durafet technology (Martz et al., 2015). The SeaFET[TM] was originally developed at the
Monterey Bay Aquarium Research Institute (Martz et al., 2010), but since has been
manufactured and distributed by Satlantic (http://www.satlantic.com), which is now incorporated
into Sea-Bird Scientific (http://www.seabird.com). The partnership between MBARI, Scripps
Institute of Oceanography, and Satlantic led the way for commercial availability of the
SeaFET[TM], providing a ready-to-deploy-factory calibration, quick start manual, and user-friendly
interface. The first generation of SeaFETs[TM] (not distributed by Sea-Bird, but by Dr. Todd Martz
at Scripps Institute of Oceanography) have been deployed in numerous field studies and were
heavily scrutinized in order to provide robust best practices for appropriate calibration and
deployment procedures (Bresnahan et al., 2014; Hofmann et al., 2011; Kapsenberg and
Hofmann, 2016; Martz et al., 2010; Matson et al., 2011; Yu et al., 2011). More recent studies
have expanded the scope of SeaFET[TM] accuracy, inter-sensor variability, operator experience,
and multi-point calibration techniques (Gonski et al., 2018; Johnson et al., 2017; Kapsenberg et
al., 2017; McLaughlin et al., 2017). Given the multitude of information regarding SeaFET[TM]
performance, coalescing all the potential sources of uncertainty in measurements (e.g., inter-
sensor variability and calibration method) can be logistically challenging for non-experienced
oceanographers who now have access to the commercially available SeaFETs[TM] distributed by
Sea-Bird.

In this study, we aimed to take advantage of two distinct coastal settings in order to
deploy and evaluate the commercially available Sea-Bird SeaFET[TM], and the potential
uncertainties that can arise with time series $pH_t$ (total scale) measurements. For this evaluation,
SeaFETs[TM] were co-deployed side-by-side to quantify inter-sensor variability, discrepancies
were examined between factory calibration, *in situ* single-point calibration, and *in situ* multi-
point calibration $pH_t$ values, and anomalous data associated with SeaFET[TM] conditioning times
were detailed and considered as potential sources of measurement inaccuracies. All evaluations
of SeaFET[TM] performance were under non-controlled source water conditions or by *in situ*
deployments. Three SeaFETs[TM] were deployed in coastal waters and were subjected to tidal
influences and freshwater input, while a fourth was compared to $pH_t$ values derived from
measurements obtained by a BoL. Finally, a spectral analysis of the quality-controlled data was
performed in order to identify the driving mechanism of $pH_t$ variability between these divergent
sites and consider possible un-accounted for calibration errors that could occur in dynamic
settings that might not be resolved using a specific calibration method.

**2 Methods**

**2.1 Apparatus: SeaFET[TM]**

The commercially available Sea-Bird SeaFET[TM] has retained the basic design of the original
SeaFET[TM] developed at MBARI (Martz et al., 2010). The SeaFET[TM] utilizes the ion sensitive
field effect transistor (ISFET) technology, and is outfitted with an internal Honeywell Durafet





and an external solid-state chloride selective electrode (Cl-ISE) along with an internal thermistor,
which derives temperature using the (Steinhart and Hart, 1968) equation. The internal reference
electrode is intrinsically insensitive to salinity over a tested range from 30 – 36 (Bresnahan et al.,
2014), with recent work even suggesting near-ideal Nernstian response to salinity as low as ~9.0
(Gonski et al., 2018). This is in converse to the chloride sensitive external electrode, which is
salinity dependent. Both electrodes demonstrate exceptional stability over a range of moderate
salinity (30 – 36) and broad temperature (-1 to 35 °C) (Bresnahan et al., 2014; Kapsenberg et al.,
2015; Martz et al., 2014, 2010). The range of salinity sensitivity for the external electrode has
even been extended down to 20, where it displays a near-ideal Nernst slope (Takeshita et al.,
2014). Sea-Bird suggests that the external reference electrode provides the more accurate and
stable $pH_t$ measurement given that chloride concentration can be precisely determined from
accurate salinity measurements. This is in agreement with previous research demonstrating that
the external electrode has a more robust stability (Martz et al., 2010). In dynamic nearshore
environments (e.g., estuaries with strong tidal and riverine fluxes), however, the $pH_t$ derived
from the internal electrode is recommended (Sea-Bird Scientific's Branham, C., pers. comm.)
despite the potential of thermodynamic hysteresis (Martz et al., 2010). Bresnahan et al. (2014)
demonstrated that the internal electrode is of the highest quality and under most scenarios
remains nearly as stable as the external electrode—this was further corroborated by Gonski et al.
(2018) with SeapHOx deployments in the Murderkill estuary, Delaware.

**2.2 Calibration**

Currently, three different calibration methods are present for the SeaFET$^{TM}$: a factory pre-
deployment single-point calibration, *in situ* single-point calibration, and an *in situ* multi-point
calibration (Bresnahan et al., 2014; Gonski et al., 2018). To properly calculate $pH_t$ from
SeaFET$^{TM}$ voltage readings, an appropriate calibration coefficient is required. The applied
calibration coefficients from the factory are a single-point, pre-deployment calibration. Given
that a conditioning period is required for the SeaFET$^{TM}$ (Bresnahan et al., 2014), these
coefficients are likely not adequate once the sensor becomes conditioned to the environment to
which it is deployed. For the internal electrode, the new calibration coefficient $k_{0i}$ can be
determined as

$$k_{0i} = -S_{Nernst} * pH_t + V_{int} - k_{2i} * T, \qquad (1)$$

and $k_{0e}$ for the external electrode

$$k_{0e} = V_{ext} - pH_t + \log\left(1 + \frac{S_t}{K_s}\right) - 2 * \log(\gamma_{HCl}) - \log(Cl_T) * S_{nernst} + k_{2e} * T \qquad (2)$$

where $V_{FET}$ is the voltage from the electrode and $k_2$ is the temperature coefficient (d$E^*$/d$T$)
applied to all SeaFETs$^{TM}$ (Martz et al., 2010). For detailed definitions of $S_{nernst}$ and the salinity
dependent constants $\gamma_{HCl}$ (HCl activity coefficient), $Cl_T$ (total chloride), $S_T$ (total sulfate), and the
$HSO_4^-$ dissociation constant $K_s$ (Dickson et al., 2007; Khoo et al., 1977) in equations 1 and 2, we
refer readers to Martz et al. (2010), Bresnahan et al. (2014), and Sea-Bird Scientific SeaFET$^{TM}$
Product Manual 2.0.0. In the literature, SeaFET$^{TM}$ calibration coefficients have been denoted as
$E^*_{int}$ and $E^*_{ext}$ (Martz et al. 2010, Bresnahan et al. 2014), however, for the purpose of this



evaluation—which specifically examines commercially available Sea-Bird SeaFETs$^{TM}$—the
adoption of $k_0$ and $k_2$ is in accordance with the preferred nomenclature from the manufacturer.
Unlike the factory pre-deployment single-point calibration, the *in situ* single-point
calibration occurs after the sensor has been deployed in the field. At the operator's discretion, a
discrete sample will be collected in direct proximity to the deployed SeaFET$^{TM}$ at the same time
that the sensor is actively making a measurement, and then measured for pH$_t$ at *in situ*
temperature and salinity. The known pH$_t$ would then be used in the above equations as the "pH$_t$"
variable. Similar to the single-point *in situ* calibration, the multi-point calibration derives a series
of calibration coefficients over a short period of time that is long enough to capture environment
variability such as tidal fluxes, and then a single calibration coefficient is averaged. Both single-
point calibration methods—pre-deployment and *in situ*—appear to be suitable for fairly static
environmental conditions, whereas the multi-point *in situ* calibration is best suited for dynamic
nearshore environments (Bresnahan et al., 2014; Gonski et al., 2018).

**2.3 SeaFET$^{TM}$ conditioning: test tank deployments**

A series of three separate test tank deployments for three SeaFETs$^{TM}$$_{395, 396, 397}$ were conducted in
order to determine the conditioning period for each sensor. Initial sensor deployments took place
in October 2016 at the Alutiiq Pride Shellfish Hatchery (APSH) in Seward, Alaska. Sensors were
deployed for a duration of 72 hours in a flow-through 60 L tank where seawater taken from a
depth of ~75 m in Resurrection Bay was sand-filtered, UV treated, and finally run through a 5
µm mesh. All three sensors were programmed with identical sampling settings (Table 1). The
onboard internal thermistor was used to calculate temperature, and measurements of seawater
salinity incoming to the hatchery were collected by a Sea-Bird Scientific SBE 45 MicroTSG
Thermosalinograph that is paired with the BoL and are available on the Alaska Ocean Observing
System (http://portal.aoos.org/real-time-sensors.php#map). Factory calibration coefficients for
the internal ($k_{0i}$, $k_{2i}$) and external ($k_{0e}$, $k_{2e}$) electrodes were retained when processing raw voltage
data.
A second tank deployment for the same three SeaFETs$^{TM}$$_{395, 396, 397}$ were deployed at the
University of Alaska, Fairbanks, in the Ocean Acidification Research Center (OARC). Seawater
collected from the APSH was delivered to the OARC test tank, ~370 L in a half-filled tank.
Seawater in the tank was circulated continuously and covered to aid in the prevention of
evaporation and photosynthesis. A co-deployed Sea-Bird SBE 16plusV2 SeaCAT (recently
serviced by Sea-Bird) collected temperature and salinity readings every 5 minutes.
SeaFETs$^{TM}$$_{395, 396, 397}$ were deployed for a duration of nine days in continuous operation mode
which forgoes the ability to set frames per burst; average number of reads was identical between
all sensors (Table 1). From 1 – 4 November 2016, duplicate discrete bottle samples were
collected in 250 ml glass bottles with screw caps at ~00:00 and 17:00 UTC per day. Bottle
samples were preserved with 20 µl of saturated HgCl$_2$ and processed at a later date for TCO$_2$ and
TA with a VINDTA 3C (Versatile Instrument for the Determination of total inorganic carbon
and titration alkalinity). The VINDTA 3C has an uncertainty typically near 0.05% (Mathis et al.,
2014, 2015a). Bottle sample pH$_t$ was calculated using CO2SYS with known TCO$_2$ and TA using
the constants provided by (Uppström, 1974) and (Lueker et al., 2000); derived pH$_t$ was then
compared against SeaFET$^{TM}$ sensor pH$_t$ to test the accuracy of both internal and external



electrodes, assuming the discrete bottle samples were the "true pH" of the seawater. Upon
recovery, all SeaFETs$^{TM}_{395, 396, 397}$ were placed into polled mode and stored with wet caps filled
with tris buffer (salinity 34, pH 8.09 at room temperature, 25 °C). Again, the factory calibration
coefficients for the internal and external electrodes were retained when raw voltage was
processed. Since the SBE 16plusV2 sampled every 5 min, salinity and temperature measured by
the SBE at each 5-minute point was repeated for the following 4 minutes in order to calculate
continuous minute readings by SeaFETs$^{TM}_{395, 396, 397}$.
A final test tank deployment of the SeaFETs$^{TM}_{395, 396, 397}$ at OARC was conducted after an
assumed adequate conditioning period of nine days (first OARC deployment). All three
SeaFETs$^{TM}_{395, 396, 397}$ had been set to polled mode after the end of the previous deployment and,
therefore, were sleeping for 83 days until this final seven day deployment. The sampling settings
were identical to the first OARC deployment for all three SeaFETs$^{TM}_{395, 396, 397}$ (Table 1). Similar
to the previous OARC tank deployment, a co-deployed Sea-Bird SBE 16plusV2 SeaCAT
collected temperature and salinity mirroring the SeaFET sampling interval of 3 hrs.
The internal thermistor of each SeaFET$^{TM}_{395, 396, 397}$ was tested for accuracy by comparing
its derived *in situ* temperature to that collected by the Sea-Bird SBE 16plusV2 during the test
tank deployments. The temperature difference between the internal thermistor and the SBE
16plusV2 was used to calculate the average and maximum discrepancy between the two
temperature readings. The temperature discrepancy was then applied to a combination of TA:
TCO$_2$ ratios over a range of salinity (20 – 35) in CO2SYS (constants: Uppström, 1974; Lueker et
al., 2000), which produced two different pH$_t$ values. The difference between these two pH$_t$
values were, therefore, concluded to be a result of the temperature discrepancy.
**2.4 SeaFET$^{TM}$ performance: field deployments**
In late winter 2017—32 days post final tank deployment—SeaFET$^{TM}_{397}$ was deployed at the
APSH and the two remaining sensors (SeaFET$^{TM}_{395, 396}$) in Kasitsna Bay within greater
Kachemak Bay, Alaska (Fig. 1). At the APSH (60° 5' 55.59"N, 149° 26' 39.80"W), incoming
seawater from Resurrection Bay at a depth of 75 m is split before running through a series of
hatchery water filters so that an unfiltered line is run directly to the BoL. The incoming line to
the BoL was then split to feed an ~11.5 L conical tank housing the SeaFET$^{TM}_{397}$ fit with the
copper bio-fouling guard; tank residence time was ~7.5 min. The SeaFET$^{TM}_{397}$ at this location
was deployed on 6 March 2017 with a robust sampling setting (Table 1). Two calibration
methods were applied for this SeaFET$^{TM}_{397}$, an *in situ* single-point calibration and an *in situ*
multi-point calibration. Both calibrations were performed 50 days after deployment on 25 April
2017 once the BoL had completed service maintenance. The single-point *in situ* calibration was
taken during midday tide transition in Resurrection Bay, while the multi-point *in situ* approach
used five (sensor sampling 3 h intervals) time points spanning an entire tidal cycle. The single-
point *in situ* calibration was used to derive $k_{0i}$ for the internal electrode (eq.1) and $k_{0e}$ for the
external electrode (eq. 2). The multi-point *in situ* calibration followed the same formulations
with the difference being the final calibration coefficient calculated was the average of the five
independently calculated calibration coefficients. Three final pH$_t$ values for the SeaFET$^{TM}_{397}$
were, therefore, calculated based upon the different calibration coefficients (factory, single-point
and multi-point *in situ* calibration) and compared against the pH$_t$ determined from continuous



$pCO_2$ measurements by the BoL and derived TA (TA-S equation, Evans et al. 2015) using
CO2SYS with constants provided by Uppström (1974) and Lueker et al. (2000). $pH_t$ uncertainty
from the BoL using this combination of measured and derived parameters is 0.007 units based on
propagating the error of the BoL $pCO_2$ uncertainty reported above with the RMSE (17 μmol kg$^{-1}$
) of the regional TA-S relationship (Orr, et al., *in prep*).
Inter-sensor variability was examined between two SeaFETs$^{TM}_{395, 396}$ deployed off the
pier at the Kasitsna Bay laboratory in Kachemak Bay (59°28' 6.71"N, 151°33'11.12"W) ~1.5 m
from the bottom: depth at this location fluctuates between ~7.5 – 16.8 m (Fig. 1). On 18 March
2017—44 days post final tank deployment—SeaFETs$^{TM}_{395, 396}$ were attached to the pier piling
directly beside one another on a single mooring frame. Both SeaFETs$^{TM}$ were wrapped with pipe
tape to minimize biofouling and fit with their respective copper biofouling guards which had a
tributyltin plug attached to the inside of the guard. The sampling settings for both SeaFETs$^{TM}_{395,}$
$_{396}$ were identical to the one at the APSH (Table 1). Five discrete reference samples were taken in
duplicate: one sample on day of deployment (UTC: 3-18-17, 18:00), two samples 1-day post-
deployment (UTC: 19 March 2017, 03:00 and 15:00), and two samples 2- and 1-day pre-
recovery of the SeaFETs$^{TM}_{395, 396}$ (UTC 3 June 2017, 03:00; 6 June 2017, 03:00). Reference
samples were collected within 30 s of the instrument sampling time period via a diver's hand
Niskin, measured for temperature and salinity with a YSI 3100 conductivity instrument, stored in
250 ml glass bottles with screw caps, poisoned with 100 μl of saturated $HgCl_2$, and secured with
teflon tape around the bottleneck threading and Parafilm wrapped on the outside of the cap.
Calibration samples were processed for $TCO_2$ and TA with a VINDTA 3C and $pH_t$ calculated
using CO2SYS with the constants provided by Uppström (1974) and Lueker et al. (2000).
Salinity measurements collected by the Kachemak Bay National Estuarine Research Reserve
data sonde, 10 km SE of the deployed sensors (59°26' 26.87"N, 151°43'15.21"W), were used
along with the SeaFET's$^{TM}$ internal thermistor readings to calculate $pH_t$ from the raw voltage
data in order to capture representative environmental conditions providing relevance for the $pH_t$
time series in this location. A static salinity of 32 was also used for all calculations of $pH_t$ as an
assessment of variability due to salinity measured from a data sonde 10 km away. A total of four
different $pH_t$ values for both SeaFETs$^{TM}_{395, 396}$ were calculated based on calibration method
(factory pre-deployment single-point calibration and the *in situ* single-point) and conditioning:
either conditioned or non-conditioned to the environment. All calculated $pH_t$ values from the
SeaFETs$^{TM}_{395, 396}$ were then compared against the remaining discrete reference bottle samples
not used for calibration. This was done in order to examine the accuracy and inter-sensor
variability difference between conditioned and non-conditioned to the environment electrodes.
Because the Kachemak Bay data sonde was located 10 km from the deployed SeaFETs$^{TM}_{395, 396}$,
the measured temperature and salinity from the discrete reference samples were used to
determine $pH_t$ for the internal and external electrodes at those specific time points. That is,
sensor accuracy for these two SeaFETs$^{TM}_{395, 396}$ was only assessed with accurate temperature and
salinity values determined from the discrete bottle samples.
A fourth SeaFET$^{TM}_{268}$ operated by the Hakai Institute was deployed on Environment
Canada's Sentry Shoal weather buoy in the Northern Strait of Georgia, BC, Canada: 49° 54'
24.00"N, 124° 59' 5.99"W (Fig.1). The Sentry Shoal mooring site is in a water depth of 15 m and
the SeaFET$^{TM}_{268}$ was affixed at a depth of 1 m. A pre-deployment bucket test was conducted for
24 h at a sampling interval of 30 min with an average of 10 samples per frame and 30 frames per



burst from 28 – 29 June 2016. SeaFET$^{TM}_{268}$ was outfitted with a copper housing guard and
wrapped with copper tape. Sensor underwent two separate deployments, an initial deployment,
and a redeployment (6 July and 27 August 2016) that occurred after the sensor was retrieved for
cleaning and maintenance. Two separate calibration samples (taken in triplicate) were taken in
accordance with each deployment, and occurred 13 and 7 days after each deployment (19 July
and 2 September 2016). For each deployment, SeaFET$^{TM}_{268}$ settings were similar to the others at
the APSH and in Kasitsna Bay (Table 1). All calibration samples were taken in triplicate at a
depth of 1 m via CTD and Niskin bottle castings and collected in 350 ml amber glass bottles with
polyurethane-lined crimp-sealed metal caps and poisoned with 200 µl of saturated $HgCl_2$, and
then processed for $TCO_2$ and $pCO_2$ with a BoL at the Hakai Institute's Quadra Island Field
Station. The measured values were used to derive $pH_t$ using CO2SYS with the constants
provided by (Uppström, 1974) and (Lueker et al., 2000) in order to perform a single-point *in situ*
calibration. Uncertainty in pH determinations from BoL $pCO_2$ and $TCO_2$ measurements was
0.006 units. After SeaFET$^{TM}_{268}$ deployment and calibration, a total of three, triplicate, reference
sample sets were taken and processed for $pH_t$ following the procedure used for calibration
samples, then compared against SeaFET $pH_t$.
**2.5 Quantifying $pH_t$ and intrinsic sensor uncertainties**

Calculating $pH_t$ from the SeaFET's$^{TM}$ raw voltage reading is dependent on temperature, salinity
and an ideal 100% Nernstian response. The software application SeaFETcom permits the
operator to automatically calculate $pH_t$ by assigning the calibration coefficient either written to
the sensor's header file or the one provided on the CD-ROM (these should be identical).
Determination of final $pH_t$ values from the first test tank deployment at the APSH were
calculated by two different operators and two sources for the factory pre-deployment single-point
calibration coefficients: header file and CD-ROM disc file. Aside from that exception, all other
final $pH_t$ values for the internal and external electrodes were calculated with the Mathworks
software MATLAB (V. 2016a) and Microsoft excel (v. 2016) using the following equations for
the internal electrode

$$pH_{int} = \frac{V_{FET|INT} - k_{0i} - k_{2i}*T}{S_{nernst}},$$   (3)
and the external electrode
$$pH_{ext} = \frac{V_{FET|EXT} - k_{0e} - k_{2e}*T}{S_{nernst}} + \log(Cl_T) + 2*\log(\gamma_{HCl}) - \log\left(1 + \frac{S_t}{K_s}\right)$$   (4)
where $V_{FET}$ is the voltage from the electrode and $k_2$ is the temperature coefficient ($dE^*/dT$)
applied to all SeaFETs$^{TM}$ (Martz et al. 2010). Again, for detailed definitions of $S_{nernst}$ and the
salinity dependent constants $\gamma_{HCl}$ (HCl activity coefficient), $Cl_T$ (total chloride), $S_T$ (total sulfate),
and the $HSO_4^-$ dissociation constant $K_s$ (Khoo et al. 1977, Dickson et al. 2007) in equations 3 and
4, we refer readers to Martz et al. (2010), Bresnahan et al. (2014), and Sea-Bird Scientific
SeaFET$^{TM}$ Product Manual 2.0.0.
**2.5.1 Sensor uncertainty**
The overall accuracy (i.e., integrated uncertainties) of every SeaFET$^{TM}$ sensor was evaluated by



quantifying all sources of potential uncertainty when calculating a final $pH_t$ from the SeaFET$^{TM}$.
The $pH_t$ uncertainty introduced by calibration method was calculated as the absolute difference
between the "true $pH_t$" and the final sensor $pH_t$ derived from either factory calibration, the
single-point *in situ* calibration, or multi-point *in situ* calibration. The "true $pH_t$" was calculated
using CO2SYS dissociation constants by Lueker et al., (2000) and Uppström, (1974) with
measured $TCO_2$ and TA via the VINDTA 3C, $TCO_2$ and $pCO_2$ measured by the BoL for discrete
samples (e.g., SeaFET$^{TM}_{268}$), and $pCO_2$ and TA (TA-S equation, Evans et al. 2015) for
continuous samples (SeaFET$^{TM}_{397}$). A one-way analysis of variance (ANOVA) and the root
mean square error (RMSE) were run and calculated in order to compare the $pH_t$ values from both
electrodes on SeaFET$^{TM}_{397}$ across calibration methods against the $pH_t$ values from the BoL. The
BoL at the APSH sampled every 5 min which produced 256 comparable sample points with a
time alignment disparity that ranged from $0 - 120$ s against SeaFET$^{TM}_{397}$. The potential $pH_t$
uncertainty based on the thermistor was calculated by using the absolute difference between the
thermistor derived temperature and that measured by the SBE 16plusV2 ($T_{diff}$) from the OARC
test tank deployments and the Kasitsna Bay SeaFETs$^{TM}_{395, 396}$ against the Seldovia data sonde 10
km away. Finally, an average inter-sensor variability uncertainty term was calculated as the
difference between the two SeaFETs$^{TM}_{395, 396}$ deployed side-by-side in Kasitsna Bay after a
single-point *in situ* calibration was performed. All uncertainty terms were calculated and collated
based on our evaluations from the Alaska deployed SeaFETs$^{TM}_{395, 396, 397}$, while SeaFET$^{TM}_{268}$
deployed at Sentry Shoal was only included when determining the accuracy uncertainty term.
Due to the disparity between reference samples for the Kasitsna Bay SeaFETs$^{TM}_{395, 396}$ and
Sentry Shoal SeaFET$^{TM}_{268}$ (two discrete reference samples) to that at the ASPH SeaFET$^{TM}_{397}$
(256 reference samples), only the average calculated difference (SeaFET$^{TM}$ $pH_t$ – "true $pH_t$") for
each calibration method and electrode was used from the APSH SeaFET$^{TM}_{397}$ and then collated
with the other reference points from the Kasitsna Bay and Sentry Shoal SeaFETs$^{TM}_{395, 396, 268}$.
**2.5.2 $pH_t$ time series analysis**
Final time series analysis was examined in the time and frequency domain using the Mathworks
software MATLAB (V. 2016a). Power spectral density was determined via Welch's method
using the pwelch function in MATLAB. Time series data was resampled and linearly
interpolated in order to compensate for the missing data points that occurred when sensors
arbitrarily stopped sampling.
**3 Results**
**3.1 Test tank and field conditions**
Finalized (i.e., calibrated) $pH_t$ values from the first test tank deployment produced two different
values, of which each was dependent on whether the calibration coefficient from the header file
or the disc file was selected, the result was a difference of ~0.0011 units for both the internal and
external electrodes. Because sensors were stored in tris buffer that lacked the addition of bromide
between tank deployments and before field deployments, an environmental conditioning period
was required for each of the Alaska SeaFETs$^{TM}_{395, 396, 397}$ once submerged in their respective
field sites. Thus, any determination of SeaFET$^{TM}$ $pH_t$ accuracy and conditioning period from



tank deployments were inconclusive and will not be considered henceforth. No SeaFETs[TM][395, 396,]
[397, 268] displayed signs of biofouling or low battery power upon recovery.
SeaFET[TM][397] deployed in parallel with the BoL at the APSH experienced a tank failure
on 8 April 2017 resulting in the sensor's emergence for 24 h. In addition, missing temperature
and salinity values resulted in gaps of $pH_t$ measurements over the entire deployment. The BoL
experienced flow control issues when initial deployment occurred on 6 March 2017 and was not
online until 18 April 2017 but, then, operated nearly consistently until 24 May 2017. All $pH_t$ and
temperature comparisons were, therefore, made beginning on 18 April 2017.
Due to the *in situ* environmental conditioning period of the Kasitsna Bay SeaFETs[TM][395,]
[396,] calibration was performed using the initial reference sample collected on 18 March 2017,
03:00 UTC and again with the reference sample collected on 3 June 2017, 03:00 UTC. Due to
high variance between duplicate reference samples (SD: 0.08 $pH_t$) on 19 March 2017, 15:00
UTC, this reference was discarded and not used for comparison or calibration. The Sentry Shoal
SeaFET[TM][268] underwent one maintenance and cleaning procedure, including a battery change,
during the ~5-month deployment (Table 1). One calibration sample (19 July 2016) and one
reference sample (9 November 2016) were averaged from duplicate rather than triplicate
replicates due to large variance from one of the replicate samples. The reference sample taken on
23 August 2016, 17:00 UTC was discarded as temperature and salinity data were missing and
SeaFET[TM][268] $pH_t$ could not be calculated. The final reference sample (UTC: 9 November 2016,
17:05) was taken 5 min after SeaFET[TM][268] sampled on 9 November 2016, 17:00 UTC.
**3.2 Thermistor response: test tank deployment**
The internal thermistor amongst the SeaFETs[TM][395, 396, 397] had a difference of less than 0.2 °C
over the entirety of the second and third tank deployments. All thermistor derived temperature
values had good alignment with the SBE 16plusV2 temperature, and consistently recorded a
slightly higher temperature. The discrepancy between the thermistor temperature and
SBE16plusV2 was minimal, and reached a maximum of 0.378 (logged by SeaFET[TM][395]) during
any time over all tank deployments. The average discrepancy, however, was ~0.21 °C when
averaging across all SeaFETs[TM][395, 396, 397] and all times.
**3.3 Field performance**
SeaFET[TM][397] deployed alongside the BoL appeared stable throughout its entire deployment and
tracked the $pH_t$ derived from the BoL well (Fig. 2). Errant spikes were present from both
electrodes throughout periods before 18 April 2017, which were a result of plumbing changes
that occurred to the APSH incoming seawater. On 10 April 2017 the internal thermistor, BoL
temp, and BoL salinity fluctuated by 3 °C and 14, respectively, over a 12 h period. These
anomalies were removed from analysis. Salinity remained relatively stable throughout the rest of
the deployment and ranged from 30.0 – 32.1. The $pH_t$ uncertainty (SeaFET[TM] – "true" $pH_t$)
decreased, and the accuracy of the SeaFET's[TM][397] internal electrode improved once the *in situ*
single-point and multi-point calibrations were performed with a RMSE decreasing from 0.5455
$pH_t$ units under factory calibration, 0.0361 $pH_t$ units for *in situ* single-point calibration and
0.0273 $pH_t$ units for the *in situ* multi-point calibration. The external electrode also improved



accuracy with *in situ* single-point and multi-point calibrations with an RMSE of 0.1077 under
factory calibration, 0.0390 for *in situ* single-point calibration and 0.0388 for the *in situ* multi-
point calibration (Fig. 2). There was a significant difference in the reduction of the $pH_t$
uncertainty for both the internal and external electrodes when utilizing *in situ* single-point and
multi-point calibration coefficients compared to the factory calibration coefficients (Table 2). In
addition, there was a significant decrease in the $pH_t$ uncertainty when using the *in situ* multi-
point calibration coefficients rather than the *in situ* single-point method for the internal electrode,
but not for the external electrode (Table 2). The $pH_t$ uncertainty of the internal electrode
decreased from 0.0294 units with an *in situ* single-point calibration to 0.0224 units after an *in
situ* multi-point calibration. It should be noted that the time alignment disparity which ranged
from 0 – 120 s is not considered a significant source of discrepancy as only 4 sample points out
of the 256 comparable points were > 0.03 units (i.e., only 4 comparable points greater than the
average $pH_t$ uncertainty found after calibration) between any one 5 min sample taken by the
BoL. The internal thermistor of SeaFET$^{TM}_{397}$ tracked the recorded BoL temperature trend fairly
(Fig. 3), but had a greater magnitude discrepancy than its test tank deployment (~0.21 °C). On
average, the thermistor temperature had an absolute difference of 2.83 °C (SD 0.35) from 18
April 2017 – 6 June 2017, which would result in a $pH_t$ uncertainty of ~0.044 units. SeaFET$^{TM}_{397}$
was not fully submerged in the conical tank leaving the top portion susceptible to air temperature
fluctuations which could have affected the thermistor readings.
The SeaFETs$^{TM}_{395, 396}$ in Kasitsna Bay improved their accuracy after an *in situ* single-
point calibration was performed (Fig. 4), however, this was only the case when sensors were not
conditioned as calibration performed after the conditioning period reduced accuracy (Fig. 5). It
should be noted that only the $pH_t$ recorded by both SeaFETs$^{TM}_{395, 396}$ at times of the reference
samples had precise salinity and temperature (temperature and salinity recorded with reference
sample rather than thermistor derived temperature) measurements as all other measurements
were calculated from salinity measured by the data sonde 10 km away, and with temperature
derived from the onboard thermistor. The $pH_t$ recorded by the external electrode at a fixed
salinity displayed little to no variance relative to $pH_t$ calculated with data sonde salinity (< 0.02
$pH_t$ difference: average whether conditioned or non-conditioned to environment). The average
$pH_t$ uncertainty from both SeaFETs$^{TM}_{395, 396}$ reduced by approximately half for the internal
electrode when not conditioned to the environment after an *in situ* single-point calibration was
performed (0.1072 and 0.1394 to 0.0475 and 0.0741 units, respectively), while the external
electrode improved only minimally from 0.0988 and 0.0963 to 0.0610 and 0.0894 units,
respectively (Fig. 4). When *in situ* single-point calibration was performed after the
SeaFETs$^{TM}_{395, 396}$ were conditioned (i.e., calibrated with reference sample taken on 4 June 2017,
03:00 UTC), the $pH_t$ uncertainty for the internal electrode reduced only minimally from factory
calibration: 0.1072 and 0.1394 to 0.0896 and 0.1240 units, respectively (Fig. 5a, b). Conversely,
the $pH_t$ error for the external electrode increased from 0.0988 and 0.0963 to 0.1011 and 0.1480,
respectively (Fig 5c, d).
Both SeaFETs$^{TM}_{395, 396}$ displayed low inter-sensor variability for the internal electrode,
and high for the external electrode after *in situ* single-point calibration was performed on sensors
not conditioned to the environment (Fig. 6, gray circles). The mean anomaly between both
SeaFET's$^{TM}_{395, 396}$ internal electrodes was 0.0525 units, whereas the external mean anomaly was
0.145 units. When measurements taken before the sensor was conditioned to the environment





(blue shaded region Fig. 6) were removed from analysis, the mean anomaly changed by < 0.006
units for both electrodes. Inter-sensor variability for both electrodes once conditioned, and after
*in situ* single-point calibration, was < 0.05 units: 0.0409 and 0.0461 units for the internal and
external electrodes, respectively (Fig. 6, black circles). When measurements recorded before the
sensors were conditioned to the environment were removed (blue shaded region Fig. 10), the
anomaly decreased further, < 0.015 units for both electrodes.
Thermistor readings on both SeaFETs$^{TM}_{395, 396}$ tracked the temperature at the Seldovia
site well, however errant spikes occurred around 18 April 2017 and again around 10 May 2017,
and continued till the end of the deployment (Fig. 7). The absolute average difference between
the thermistor values and the Seldovia data sonde was 0.281 °C (SD 0.295), nearly identical to
the difference displayed during the test tank deployments, average 0.21 °C.
At Sentry Shoal, temperature and salinity seasonally fluctuated and ranged from 8.71 –
21.8 °C and 23.4 – 29.4, respectively. There was no clear distinction in greater accuracy between
the internal and external electrodes after *in situ* single-point calibration was performed. While the
external electrode did display a lower $pH_t$ average uncertainty, this was based on only two
reference points, one of which had a time discrepancy of 5 min (9 November 2016, 17:05 UTC).
Only two reference samples were comparable against SeaFET$^{TM}_{268}$ $pH_t$ due to the loss of salinity
and temperature data on 23 August 2016, 17:00 UTC. Reference samples on 26 September 2016
and 9 November 2016 were, therefore, compared using the new calibration coefficients
determined after redeployment on 27 August 2016. The average $pH_t$ uncertainty was < 0.0115
units for both electrodes (Fig. 8) compared to average $pH_t$ uncertainties of 0.0244 and 0.0560
units for the internal and external electrodes, respectively, if initial calibration coefficients from
19 July 2016 were retained. The low $pH_t$ uncertainty (< 0.0137 units) determined after the *in situ*
single-point calibration, however, was still greater than the average $pH_t$ uncertainty under factory
calibration: < 0.005 units for both electrodes (Fig 8).

**3.4 Spectral analysis**

All SeaFETs$^{TM}_{395, 396, 397, 268}$ displayed a mixed semi-diurnal tidal response during all field
deployments (Fig. 9). SeaFETs$^{TM}_{395, 396}$ at Kasitsna Bay had a stronger amplitude response at a
frequency of two cycles d$^{-1}$, whereas SeaFET$^{TM}_{397}$ had a greater amplitude at one cycle d$^{-1}$ (Fig.
9a, c, d). All three SeaFETs$^{TM}_{395, 396, 397}$ in Alaskan waters had a strong amplitude signal of 1
cycle every 21 days, with an addition signal of one cycle every three days for SeaFE$_T$$^{TM}_{397}$. The
amplitude signal for SeaFET$^{TM}_{397}$ shifted depending on source of measurement (BoL, internal or
external electrode), however, all measurement sources followed the same frequency pattern (Fig
9a). SeaFET$^{TM}_{268}$ displayed a strong signal at a frequency of zero as well as at one and two
cycles d$^{-1}$ (Fig 9a).

**3.5 Intrinsic uncertainty and accuracy**

Among the calculated potential sources of uncertainty in $pH_t$, inter-sensor variability (difference
between SeaFET's$^{TM}$ $pH_t$) and sensor accuracy produced the greatest uncertainty discrepancies
for the internal and external electrodes under factory calibration (Fig. 10). The $pH_t$ uncertainty
(i.e., overall sensor accuracy) for the internal electrode reduced a greater degree than the external





electrode at every ordinal calibration method: factory, *in situ* single-point, to *in situ* multi-point
calibration (Fig. 10). This was not the case for the external electrode, however, as the overall $pH_t$
accuracy was greater when factory calibration was used compared to an *in situ* single-point
calibration was performed after the sensor was conditioned. The thermistor uncertainty (i.e.,
uncertainty when calculating $pH_t$ based on the thermistor temperature rather than a more accurate
temperature gauge) produced a $pH_t$ uncertainty of 0.0044 units, and was based on the recorded
values by SeaFETs$^{TM}_{395, 396}$. Even though the temperature-derived values from the thermistor of
SeaFETs$^{TM}_{395, 396}$ were compared against a data sonde 10 km away, the average $T_{diff}$ values were
consistent with the $T_{diff}$ calculated from the test tank deployments (within 0.07°C) and, therefore,
provided an adequate resolution to determine a thermistor uncertainty value.
**4 Discussion**
Obtaining accurate and precise measurements of pH in nearshore coastal waters is crucial for
understanding changing trends, dynamics, and current baselines of acidification in these—
"susceptible to change"—marine domains. For dynamic nearshore systems, the current standard
of OA weather (carbonate chemistry variability on timescales of days to months) accuracy
should have an uncertainty no greater than 0.02 pH units according to the Global Ocean
Acidification Observing Network (Newton et al. 2015). Previous evaluations of the SeaFET$^{TM}$
sensor package have demonstrated accuracy for both electrodes to be better than 0.02 pH units,
with a range between 0.01 – 0.04 units for the internal electrode in more dynamic environments
(Bresnahan et al., 2014; Gonski, 2018; Martz et al., 2010). Based on our findings, we observed
an accuracy range of 0.009 – 0.148 $pH_t$ units after sensors were conditioned and *in situ* single-
point or multi-point calibrations were performed for the internal and external electrodes. This
range decreased when SeaFETs$^{TM}_{395, 396}$ from Kasitsna Bay were calibrated with reference
samples taken at initial deployment (i.e., non-conditioned to environment). For SeaFET$^{TM}_{397}$, the
internal electrode's accuracy was nearly identical to that of the external electrode after an *in situ*
multi-point calibration (Fig. 2), suggesting that the internal electrode can produce a highly
precise $pH_t$ measurement comparable to the BoL with an accuracy meeting the standards of the
OA weather measurements (Newton et al. 2015). This is not to suggest that the SeaFET$^{TM}$ can
replace the BoL, particularly because the BoL can capture multiple carbonate chemistry
measurements thereby fully constraining the system and identifying potential decoupling of the
carbonate system in estuarine waters (Bandstra et al., 2006; Hales et al., 2016). Nonetheless, the
SeaFET$^{TM}$ can provide an accurate measurement of $pH_t$ in nearshore waters when SeaFET$^{TM}$
operation is executed with high precision.
SeaFETs$^{TM}_{397, 268}$ deployed at the APSH and at Sentry Shoal displayed the lowest
uncertainty and greatest precision of $pH_t$ measurements (Fig. 2 and 8). In both instances, the
SeaFETs$^{TM}_{397, 268}$ were adequately conditioned (i.e., subjected to *in situ* conditions for ~50 days)
before calibration was performed. The greater overall accuracy displayed by the SeaFET$^{TM}_{268}$ at
Sentry Shoal may be due to the fact that the sensor was exposed to *in situ* conditions for a longer
period of time and re-calibrated multiple times to the same environment. Further, calibration and
reference sample $pH_t$ was derived from $TCO_2$ and $pCO_2$ processed by the BoL at Sentry Shoal
and from $pCO_2$ (also measured by BoL) and the TA-salinity relationship (Evans et al. 2015) at
the APSH. It is unclear as to why the sensor accuracy of both Kasitsna Bay SeaFETs$^{TM}_{395, 396}$
was substantially less than the SeaFETs$^{TM}_{397, 268}$ at the APSH or Sentry Shoal. A potential reason



for the low accuracy may be that sensors were calibrated at a reference point that was extreme
relative to the time series $pH_t$ signal—that is, calibrated at a time of high variability. In this case,
performing an *in situ* multiple-point calibration could have reduced the uncertainty and increased
the accuracy. While previous studies have found that collection and preservation of calibration
and reference samples can result in a decrease in accuracy depending on operator experience
(McLaughlin et al., 2017), the operator in this study was considered to have substantial
experience conducting such operations used in this evaluation. In addition, given the increased
$pH_t$ variability over a short temporal period—which can be seen at the end of the Kasitsna Bay
deployment (Fig. 4 and 5)—and the low discrepancy between duplicate reference samples, the
former reasoning (i.e., calibrated to an extreme reference point) is a more reasonable explanation
for the reduced accuracy by the Kasitsna Bay SeaFETs$^{TM}_{395, 396}$ than operator experience. We re-
iterate here that reference sample temperature and salinity were used to calculate SeaFET$^{TM}$ $pH_t$
at the time points in which sensor $pH_t$ and reference sample $pH_t$ were compared, thus salinity
was not a confounding factor.
Despite the lower accuracy of the Kasitsna Bay SeaFETs$^{TM}_{395, 396}$, the two sensors
provided a better insight of inter-sensor variability for non-conditioned to the environment and
conditioned electrodes. After *in situ* single-point calibration for conditioned sensors, the average
inter-sensor variability decreased for the internal electrode by ~80%, and >300% for the external
electrode (Fig. 6). The inter-sensor variability reported here was still greater than previous
findings (Kapsenberg et al., 2017), however, the comparison made in this study was done in the
field compared to controlled laboratory conditions as in Kapsenberg et al. (2017). And while
non-homogenized water could lead to anomalies in $pH_t$ measurements by the Kasitsna Bay
SeaFETs$^{TM}_{395, 396}$, it is unlikely that water was consistently non-homogenized over the entirety of
a deployment at a distance of < 20 cm (distance between electrodes on each SeaFET$^{TM}$).
Furthermore, due to the dynamic nature of Kachemak Bay, where the tidal exchanges are
extreme, averaging 4.73 m, it is unlikely that micro-heterogeneity of seawater is the driving force
behind the observed differences in $pH_t$ measurements that were observed between SeaFETs$^{TM}_{395,}$
$_{396}$. There was a tradeoff for a decrease in inter-sensor variability, as the *in situ* single-point
calibration performed after sensors were conditioned resulted in a decrease in accuracy compared
to an *in situ* single-point calibration performed for sensors not conditioned to the environment. It
should be noted that we do not consider salinity to be a potential source of uncertainty for inter-
sensor variability because the $pH_t$ difference using data sonde salinity compared to a fixed
salinity resulted in an anomaly of < 0.005 units.
The Sentry Shoal SeaFET$^{TM}_{268}$ had the lowest average $pH_t$ uncertainty for both electrodes
after *in situ* single-point calibration was performed, however, these were still greater than the $pH_t$
uncertainty determined using the factory calibration coefficients. This specific example
highlights two possibilities: (1) the role of inter-sensor variability, as this may be a coincidental
case given the uncertainty observed when quantifying inter-sensor variability, and (2) the
influence of variance within a calibration sample set. For the case of SeaFET$^{TM}_{268}$, the replicate
calibration samples collected on 19 July 2016 and 2 September 2016 for the first and second
deployments had standard deviations of 0.016 and 0.005 $pH_t$ units, respectively. For instances of
generally close agreement between factory and *in situ* calibrated data, the variance in the
calibration sample set may contribute to better agreement between factory calibrated sensor $pH_t$
data and average discrete sample $pH_t$ measurements. It should also be noted that pre-deployment



calibration can provide highly accurate measurements by the Honeywell Durafet (internal
electrode), however, matching exact conditions to those at the field site are necessary (Johnson et
al., 2017), and this was not likely the case for the factory provided calibration coefficients.

The evaluation of SeaFET$^{TM}$ performance presented here corroborates and contrasts with
previous studies examining the overall accuracy and precision of $pH_t$ measurements made by
these oceanographic instruments. While the accuracy of two SeaFETs$^{TM}_{397,\,268}$ fall well within
the range determined from previous studies, the accuracy of SeaFETs$^{TM}_{395,\,396}$ at Kasitsna Bay
lay outside the bounds of what has been report in the primary literature (Bresnahan et al., 2014;
Gonski et al., 2018; Johnson et al., 2017; Kapsenberg et al., 2017; Martz et al., 2010).
Nevertheless, it is relevant to report the potential uncertainties possible when operating
SeaFETs$^{TM}$ as a multitude of factors can influence the overall accuracy (e.g., operator, sample
preservation, electrode conditioning, calibration measurements), therefore, the potential
uncertainties calculated in this study represent the upper limit of an average uncertainty compiled
from four different SeaFETs$^{TM}$ (Fig. 10). The utility of such an analysis provides a confidence in
SeaFET$^{TM}$ operation, and highlights all the potential uncertainties that need to be considered
when deploying the sensors in the field. For example, we have included a thermistor uncertainty
term determined from the test tank and field deployments of the Alaska SeaFETs$^{TM}_{395,\,396,\,397,}$
even though a suitable solution around this issue would be to apply an offset to the thermistor
temperature given it was compared to more robust temperature measurements conducted before
field deployment. It should be noted, that in this case, the thermistor uncertainty observed from
SeaFET$^{TM}_{397}$ against the BoL was excluded as the lag time between thermistor response and tank
residence time likely confounded the comparison. The potential $pH_t$ uncertainties presented here
should serve as a guide for SeaFET$^{TM}$ operators in order to better understand the source of an
uncertainty and take the necessary steps to improve SeaFET$^{TM}$ measurements. Bresnahan et al.
(2014) acknowledged that relying on the SeaFET$^{TM}$ for an accurate pH measurement should be
viewed cautiously if additional biogeochemical sensors are not co-deployed to cross-validate the
stability and accuracy of the SeaFET's$^{TM}$ electrodes, therefore, being fully aware of all the
potential uncertainties presented here will only further aid SeaFET$^{TM}$ operators.

The time series data provided by the SeaFET$^{TM}$ deployments in this study have expanded
the scope of spatial $pH_t$ variability along the North American west coast. The SeaFETs$^{TM}_{395,\,396}$
deployed in Kasitsna Bay provide some of the first high temporal resolution measurements of
$pH_t$ in this region. During this spring deployment, it appears that semi-diurnal tidal fluctuations
are the dominant contributor to $pH_t$ variability with an additional cycle occurring every 21 days
coinciding with the seasonal spring and neap tides (Fig. 9). The SeaFET$^{TM}_{268}$ at Sentry Shoal
also displays a strong $pH_t$ response to the semi-diurnal mixed tidal cycle. A strong signal is also
present at a frequency of zero, and is likely a result of the long, across-season, time series. That
is, over the course of the entire deployment which went from summer into late fall, seasonal
drivers of $pH_t$ (e.g., decrease in water temperature) confounded repetitive frequency patterns. In
addition, Sentry Shoal may have a weaker tidal signature relative to other $pH_t$ modulators that do
not follow a cyclical pattern such as water mass intrusion, inconsistent metabolic cycles from the
end of summer into the fall season, and a shift to the rainy season.

As an elaboration on the power spectral density analysis, we suggest this form of
frequency analysis can be utilized to better understand the system in which a SeaFET$^{TM}$ is




deployed, thus informing the operator as to what the drivers of their system are, and when to
calibrate the sensor. It is possible that in a highly dynamic setting, the sensor could re-condition
over time periods not resolved in a multi-point calibration sampling scheme, and this could
enhance sensor inaccuracies. For example, in Kasitsna Bay, a strong semi-diurnal tide cycle was
present, so upon redeployment in this area, if possible, the best calibration approach would be an
*in situ* multi-point calibration between the M2 cycle. Alternatively, if the system is not driven by
a strong tidal signature (e.g., non-coastal region), an *in situ* single point calibration may be a
reasonable approach.

**5 Conclusion**

The following evaluation of the Sea-Bird SeaFET$^{TM}$ helped elucidate the overall
accuracy and highlighted the potential uncertainties and pitfalls of operating and obtaining $pH_t$
measurements by the internal and external electrode pair. We found that the internal electrode
provided the more robust measurement in nearshore estuarine waters when an *in situ* multi-point
calibration was performed (Fig. 10). The quantified potential $pH_t$ uncertainty is based
specifically on our findings, whereas further results may minimize this uncertainty given
additional evaluations. However, the results here provide an upper limit of the $pH_t$ uncertainty
that may be observed when operating a Sea-Bird SeaFET$^{TM}$. Further, high temporal resolution
$pH_t$ measurements in nearshore Canadian and Alaskan waters provide a better understanding of
the drivers modulating pH on short timescales. Given the application, the Sea-Bird SeaFET$^{TM}$
can provide a reliable and accurate $pH_t$ measurement which can be utilized to broaden the
coverage of understanding pH variability in nearshore and open-ocean waters.

**Acknowledgments**

The authors would like to thank Jeff Hetrick and Jacqueline Ramsey at the Alutiiq Pride
Shellfish Hatchery for providing their facilities and services for this evaluation. We would also
like to thank Angela Doroff at the Kasitsna Bay laboratory for providing facilities for SeaFET$^{TM}$
deployments. Funding for this project was provided in part by the University of Alaska
Fairbanks College of Fisheries and Ocean Sciences. WE and KP thank the Pacific Salmon
Foundation and Environment Canada for providing the platform for deploying SeaFET 268, the
University of Alaska Fairbanks Ocean Acidification Research Center for the long-term use of
SeaFET 268, and the Tula Foundation for supporting their efforts with this work.

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





**Table 1.** Deployment regime of all four SeaFETs[TM] including deployment location, date, and calibration methods performed. *Non-controlled source water pumped directly from Resurrection Bay, AK, USA.

| Location (Tank or Field) | Date | SeaFET[TM] ID | Average reads frame$^{-1}$ | Frames Burst$^{-1}$ | Sampling Freq. (min) | Calibration method |
|---|---|---|---|---|---|---|
| APSH — *Tank* | 5 – 8 October 2016 | 395, 396, 397 | 1 | 10 | 5 | Factory |
| OARC — *Tank* | 26 October – 3 November 2016 | 395, 396, 397 | 3 | — | Continuous | Factory |
| OARC — *Tank* | 26 January – 1 February 2017 | 395, 396, 397 | 1 | 10 | 180 | Factory |
| APSH *Field** | 5 March – 6 June 2017 | 397 | 10 | 30 | 180 | Factory, SP and MP *in situ* |
| Kachemak Bay *Field* | 18 March – 4 June 2017 | 395, 396 | 10 | 30 | 180 | Factory, SP *in situ* |
| Sentry Shoal *Field* | 6 July – 23 August, 27 August – 28 November 2016 | 268 | 10 | 30 | 30 | Factory, SP *in situ* |





**Table 2.** One-way Analysis of variance comparing the pH$_t$ error (SeaFET$^{TM}$ pH$_t$ − BoL pH$_t$) across calibration methods for both the internal and external electrodes onboard SeaFETs$^{TM}$$_{268}$ at Sentry Shoal (factory calibration and *in situ* single-point calibration) and SeaFET$^{TM}$$_{397}$ at the Alutiiq Pride Shellfish Hatchery (factory calibration, *in situ* single-point calibration, and *in situ* multi-point calibration). Bold type denotes statistical significance.

| Site | Electrode | Source | SS | df | MS | F | p-value |
|---|---|---|---|---|---|---|---|
| APSH | Internal | Fac Cal. Vs. Sinlge-point | 27.5 | 1 | 27.5 | 4.96E+04 | **< 0.001** |
|  |  | Error | 0.225 | 406 | 0.001 |  |  |
|  |  | Total | 27.7 | 407 |  |  |  |
| APSH | External | Fac Cal. Vs. Sinlge-point | 0.681 | 1 | 0.681 | 536 | **< 0.001** |
|  |  | Error | 0.516 | 406 | 0.001 |  |  |
|  |  | Total | 1.19 | 407 |  |  |  |
| APSH | Internal | Factory Cal. vs. Multi-point | 28.3 | 1 | 28.3 | 6.19E+04 | **< 0.001** |
|  |  | Error | 0.185 | 406 | 0.001 |  |  |
|  |  | Total | 28.5 | 407 |  |  |  |
| APSH | External | Factory Cal. vs. Multi-point | 0.692 | 1 | 0.692 | 539 | **< 0.001** |
|  |  | Error | 0.521 | 406 | 0.001 |  |  |
|  |  | Total | 1.21 | 407 |  |  |  |
| APSH | Internal | Single-point vs. Multi-point | 0.005 | 1 | 0.005 | 15.0 | **< 0.001** |
|  |  | Error | 0.143 | 406 | 0.000 |  |  |
|  |  | Total | 0.148 | 407 |  |  |  |
| APSH | External | Single-point vs. Multi-point | 0.000 | 1 | 0.000 | 0.040 | 0.843 |
|  |  | Error | 0.415 | 406 | 0.001 |  |  |
|  |  | Total | 0.415 | 407 |  |  |  |



**Figure 1.**

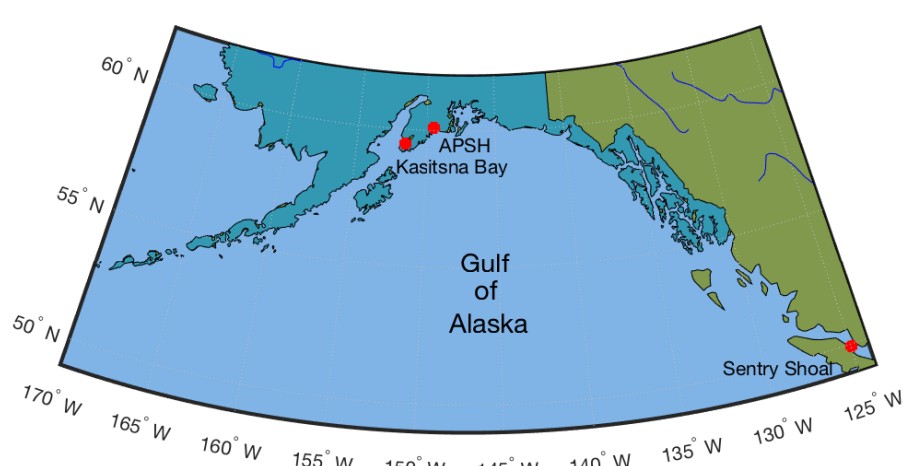

Geographical map with locations of SeaFET$^{TM}$ field deployments along Alaska's, USA, south-
central coast and one location in the Strait of Georgia, British Columbia, Canada.




**Figure 2.**

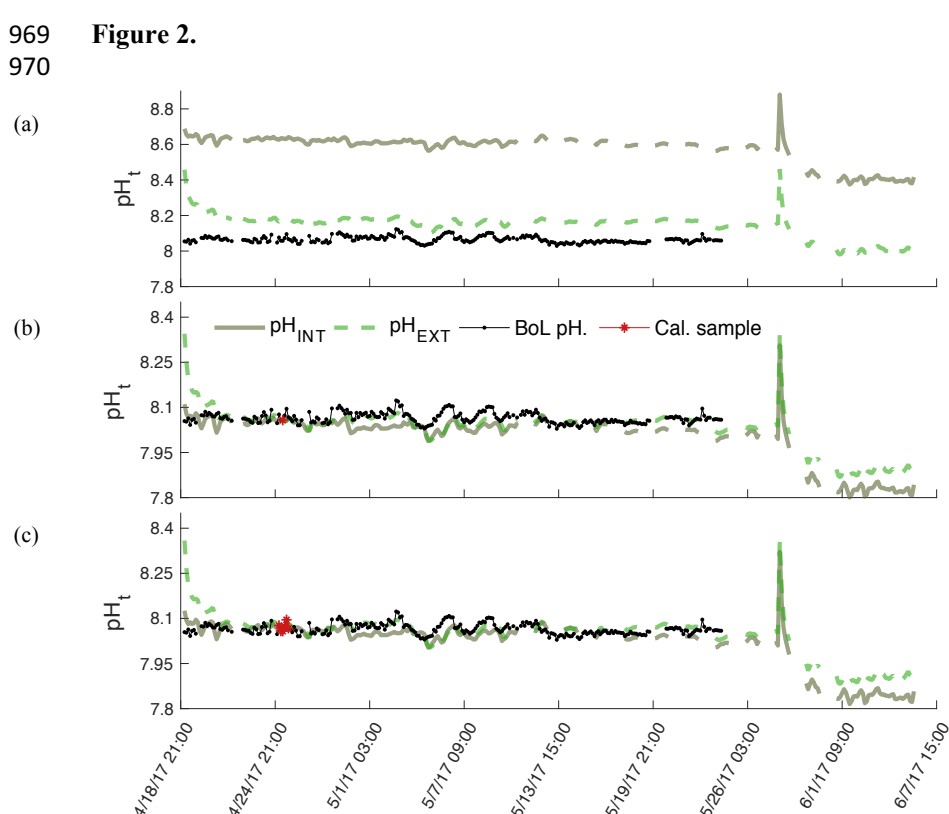

pH$_t$ recorded by the internal (solid) and external (dashed) electrodes on SeaFET$^{TM}_{397}$ deployed in
parallel with the BoL at the Alutiiq Pride Shellfish Hatchery. pH$_t$ from both electrodes is shown
when derived using factory calibration (FC) coefficients (panel a), *in situ* single-point (SC)
calibration coefficients (panel b), and *in situ* multi-point (MC) calibration coefficients (panel c).
Black solid line is pH$_t$ derived from continuous $p$CO$_2$ measurements recorded by the BoL and
derived TA from the TA-S relationship (Evans et al. 2015). Red circles are the calibration points
from the BoL data.




**Figure 3.**

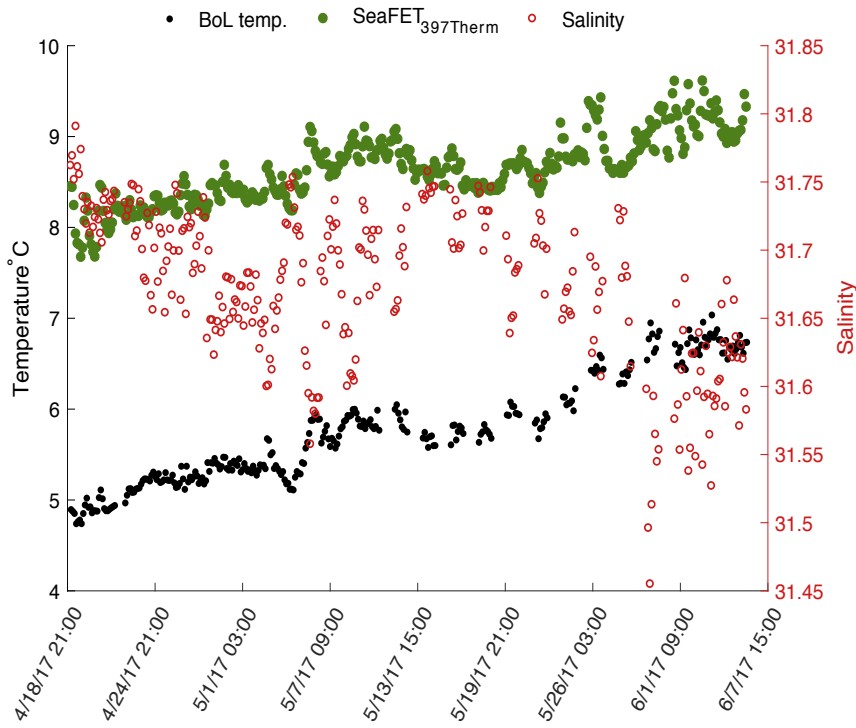

Temperature derived from the internal thermistor on SeaFET$^{TM}$$_{397}$ (green circles) and the
temperature recorded by the BoL (black circles) at the Alutiiq Pride Shellfish Hatchery from late
winter through spring 2017. Salinity (red circles) recorded by the BoL on the right y-axis.




**Figure 4.**

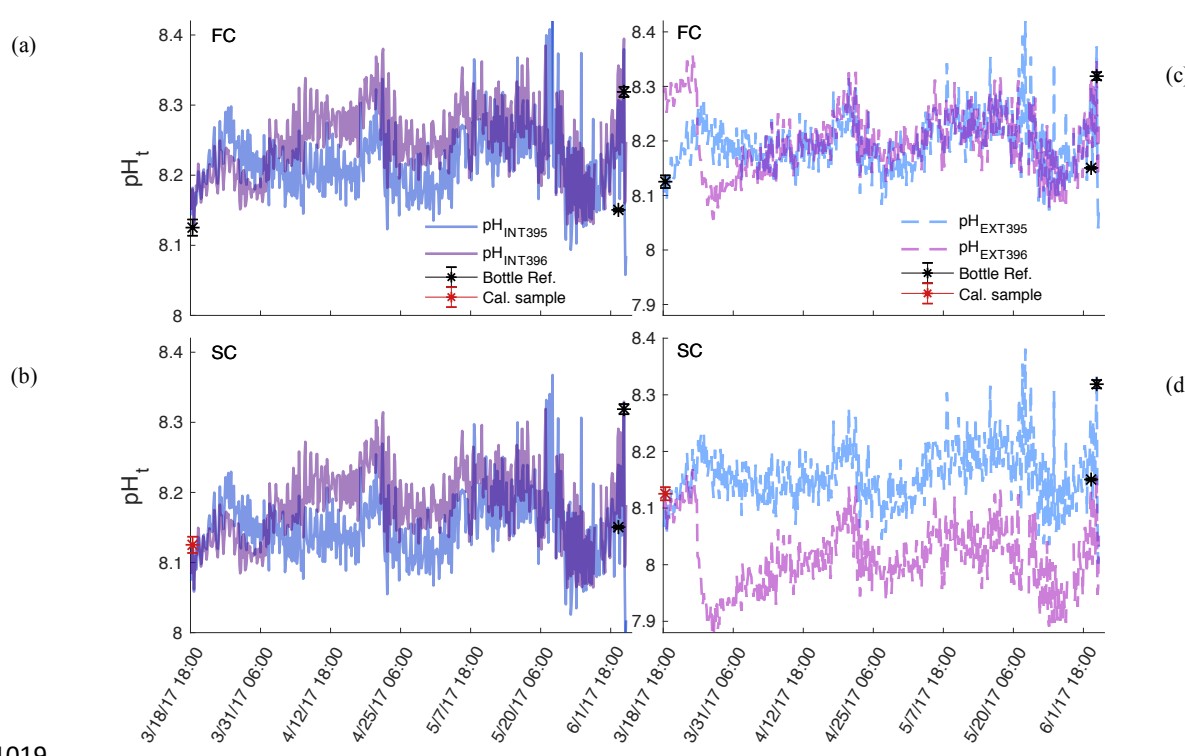

Comparison of pH$_t$ recorded by the internal (panel a and b) and external (panel c and d)
electrodes on SeaFET$^{TM}$$_{395}$ (blue) and SeaFET$^{TM}$$_{396}$ (purple) before they were conditioned to the
environment (non-conditioned) deployed in Kasitsna Bay, AK, based on calibration method:
factory calibration (FC) and *in situ* single-point (SC) calibration. Discrete reference samples
(black asterisks) and calibration sample (red asterisks) were collected 36 and 12 h pre-SeaFET$^{TM}$
recovery, and < 24 h post-deployment, respectively. Temperature and salinity measurements
collected on reference and calibration samples were used to derive SeaFET$^{TM}$ pH$_t$ at those given
time points. All other SeaFET$^{TM}$ pH$_t$ measurements use thermistor temperature and salinity
logged by Kasitsna Bay data sonde.



**Figure 5.**

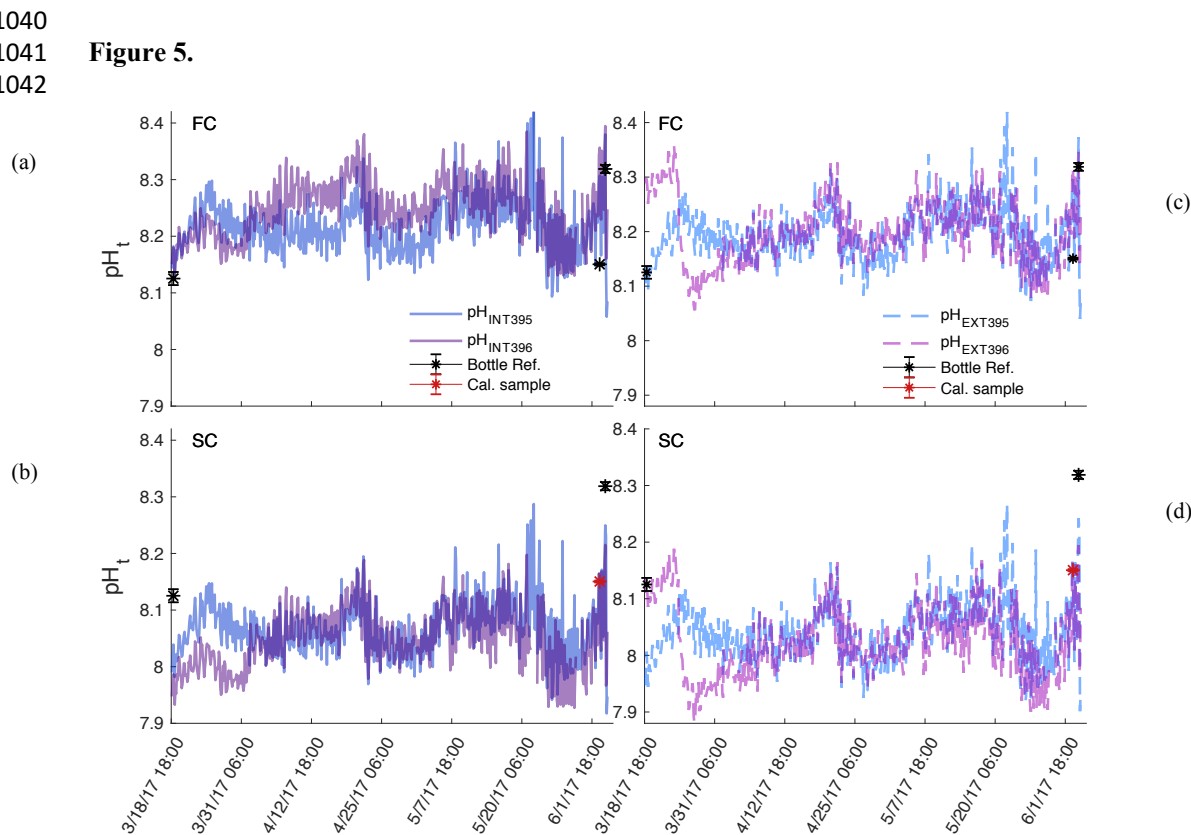

Comparison of pH$_t$ recorded by the internal (panel a and b) and external (panel c and d)
electrodes on conditioned SeaFET$^{TM}_{395}$ (blue) and SeaFET$^{TM}_{396}$ (purple) deployed in Kasitsna
Bay, AK, based on calibration method: factory calibration (FC) and *in situ* single-point (SC)
calibration. Discrete reference samples (black asterisks) and calibration sample (red asterisks)
were collected < 24 h post deployment and 12 h pre-SeaFET$^{TM}$ recovery, while calibration
sample was collected 36 h pre-SeaFET$^{TM}$ recovery. Temperature and salinity measurements
collected on reference and calibration samples were used to derive SeaFET$^{TM}$ pH$_t$ at those given
time points. All other SeaFET$^{TM}$ pH$_t$ measurements use thermistor temperature and salinity
logged by Kasitsna Bay data sonde.



**Figure 6.**

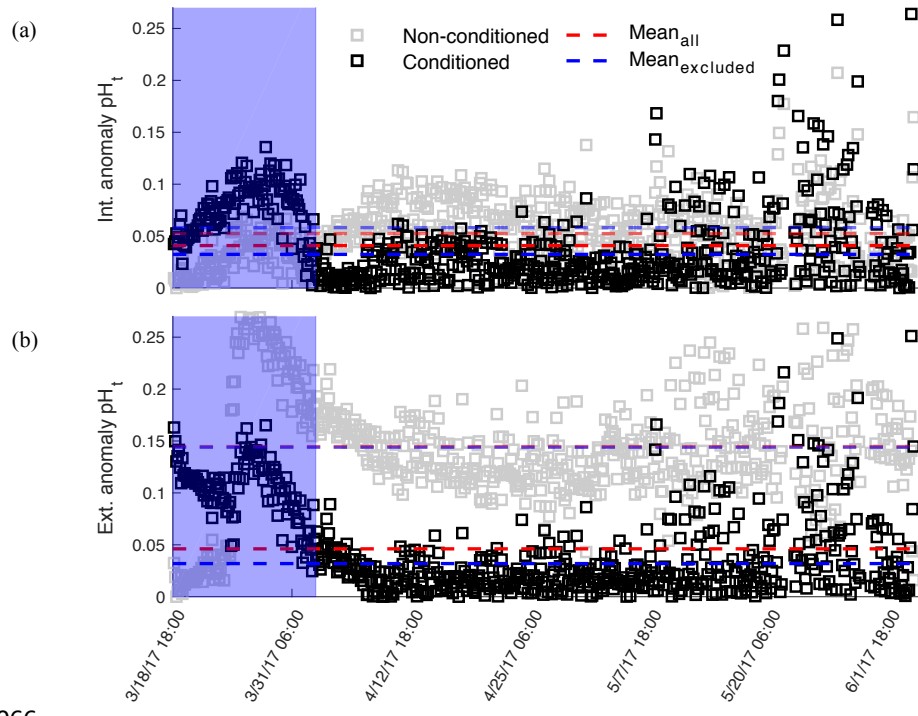

Mean $pH_t$ anomaly between *in situ* single-point calibrated SeaFET$^{TM}_{395}$ and SeaFET$^{TM}_{396}$
internal (panel a) and external (panel b) electrodes during parallel deployment in Kasitsna Bay,
AK. Intra-anomaly comparison based on calibration sample taken at initial deployment (< 24 h
non-conditioned, gray squares) and end of deployment (36 h pre-recovery, black squares).
Shaded blue region indicates conditioning period. Data points in blue region omitted when mean
anomaly was calculated (non-conditioned: transparent blue-dashed line; conditioned: bold blue-
dashed line) compared to mean anomaly from entire data set (non-conditioned to environment:
red-dashed line; conditioned: red- dashed line).



**Figure 7.**

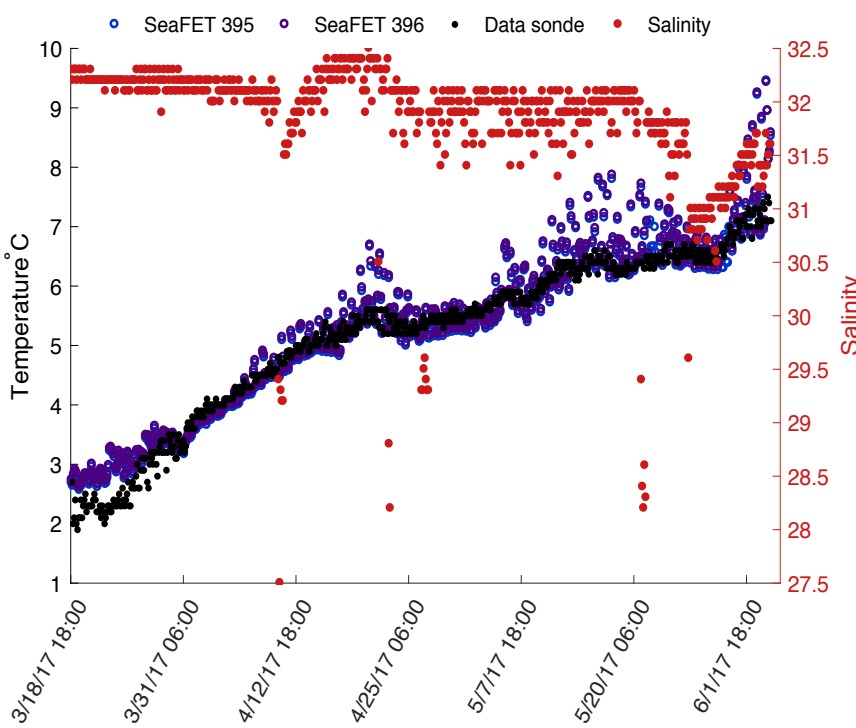

Temperature derived from the internal thermistor on SeaFET$^{TM}_{395}$ (blue) and SeaFET$^{TM}_{396}$
(purple) compared against the temperature recorded by the Kachemak Bay National Estuarine
Research Reserve data sonde. Salinity (Red circles) recorded by Kachemak Bay data sonde on
the right y-axis.

10.5194/os-2018-52
Ocean Science




**Figure 8.**

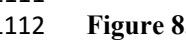
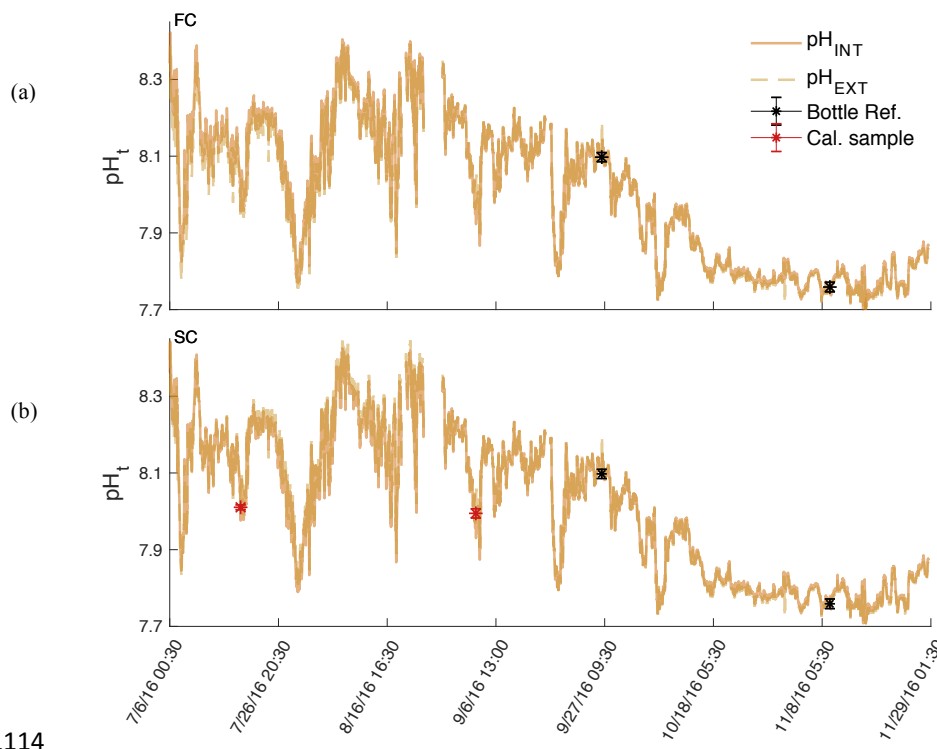

pH$_t$ recorded by the internal (solid) and external (dashed) electrodes on SeaFET$^{TM}_{268}$ deployed at
the Sentry Shoal mooring. pH$_t$ from both electrodes is shown when derived using factory
calibration (FC) coefficients (panel a) and *in situ* single-point (SC) calibration coefficients (panel
b). Black asterisks are references samples taken after initial calibration and recalibration (red
asterisk), where pH$_t$ was derived from TCO$_2$ and $p$CO$_2$ measurements made on the BoL at the
Hakai Institute's Quadra Island Field Station.



**Figure 9.**

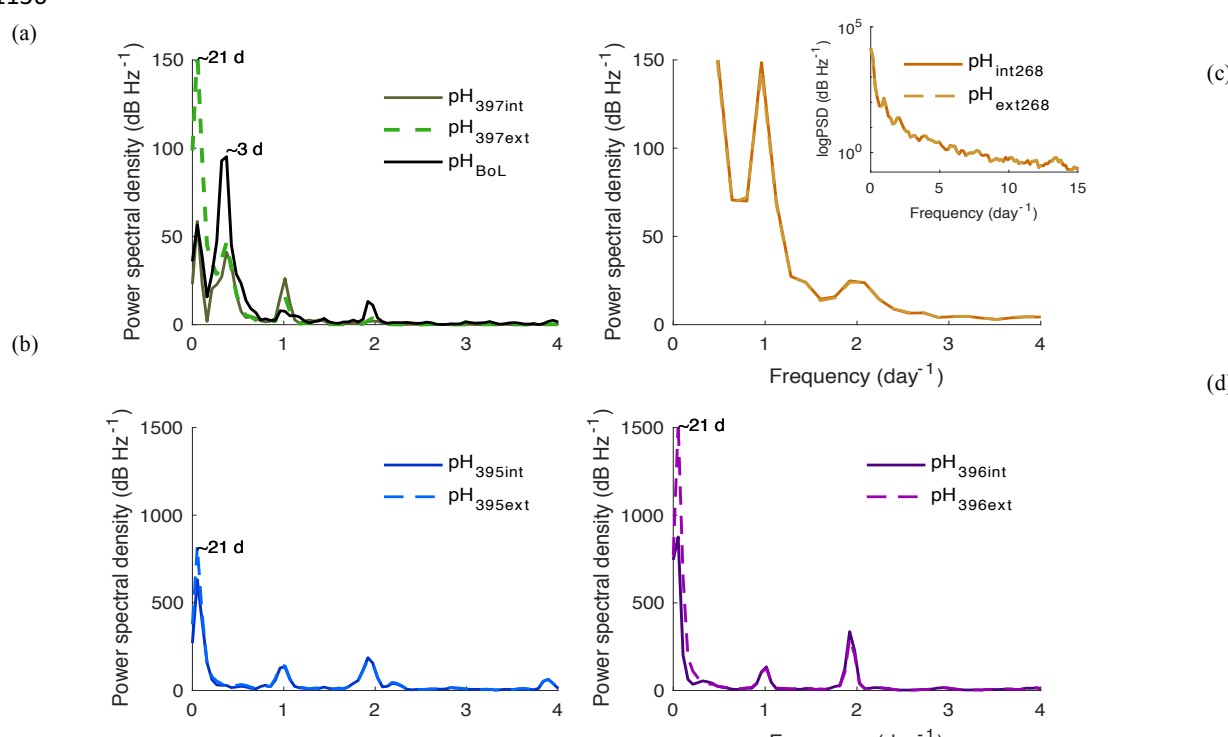

Power spectral density (PSD) analysis of pH$_t$ in frequency per day for SeaFETs$^{TM}$ 397 (panel a),
268 (panel b), 395 (panel c), and 396 (panel c). Inset in panel b is log base 10 transformed PSD
analysis of same data set. All internal electrodes marked as solid colored lines while external
electrodes are colored dashed lines. BoL data set marked as solid black line (panel a).



**Figure 10**

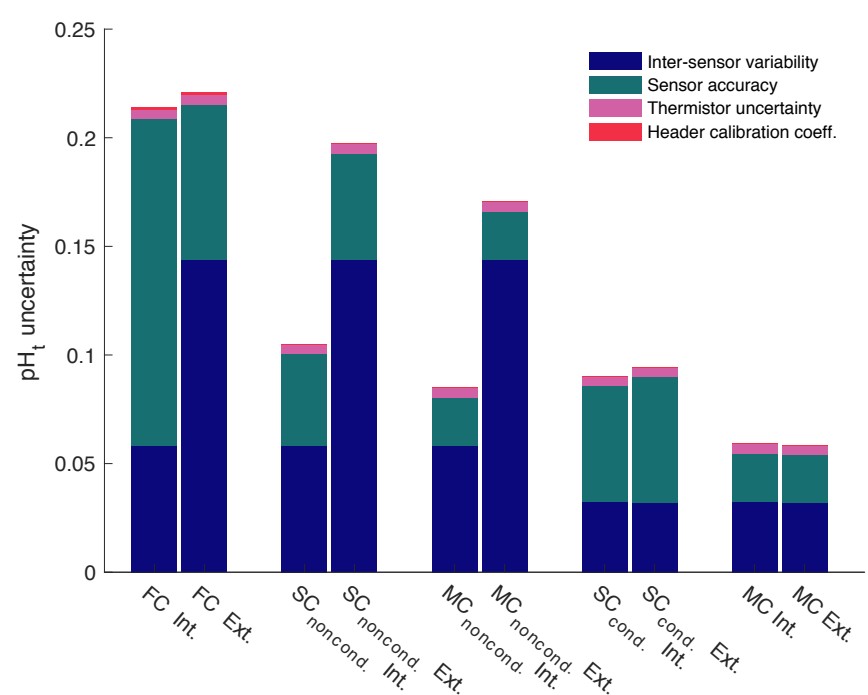

Quantified uncertainties based on field deployments of all Sea-Bird SeaFETs[TM] separated by electrode calibration method (FC: factory; SC: single-point; MC: multi-point), and calibration time for SeaFETs[TM] 395 and 396 (i.e., non-conditioned to environment and conditioned). $pH_t$ accuracy uncertainty calculated as the mean difference when comparing the absolute difference between reference samples and SeaFETs[TM] 395 (non-conditioned to environment and conditioned), 396 (non-conditioned to environment and conditioned), and 268 as well as the average absolute difference between SeaFET[TM] 397 and the BoL. Inter-sensor variability uncertainty determined by comparing SeaFETs[TM] 395 (non-conditioned to environment and conditioned) and 396 (non-conditioned to environment and conditioned), deployed side-by-side in Kasitsna Bay. Thermistor uncertainty is calculated $pH_t$ error when using thermistor derived temperature rather than external temperature sensor determined from SeaFETs[TM] 395 and 396. Header calibration coefficient uncertainty is the discrepancy in $pH_t$ when using SeaFETcom factory calibration coefficients from header file rather than disc file.