# Peer review of "An Evaluation of the Performance of Sea-Bird Scientific's SeaFETTM Autonomous pH Sensor"

_Ocean Science, 2018_

## Referee Comment (RC1) · Anonymous Referee #1 · 29 May 2018

The research presented by Miller et al. is an evaluation of the SeaFET, a seawater pH sensor with fairly common usage within the ocean carbon community. The evaluation included both tank and field testing paired with independent, high-quality pH validation measurements. Miller et al. find that for the three coastal locations utilized in this study, a multi-point calibration results in the highest accuracy for SeaFET pH. This is counter to the more commonly used procedure of keeping the factory calibration as presented by Bresnahan et al. 2014.

This paper is an important contribution to understanding how this sensor performs in the environments in which it is deployed. I recommend publication in Ocean Sciences

after the authors consider some comments below that may make the paper more relevant to other sensor users and may reduce environmental error in the sensor uncertainty estimates.

Major comments:

1) Assess uncertainty if following today's existing SOP: While this study clearly shows the utility of single- and multi-point calibrations in these three coastal systems, it would be useful to discuss the uncertainty observed when following the existing SeaFET standard operating procedure (SOP). I believe the SOP described by Bresnahan et al. 2014 is to use the factory calibration, but correct the dataset to some independent measure of "true" pH (e.g., discrete bottle samples, pH derived from other biogeochemical sensors combined with locally-constrained carbon system algorithm or TA-S relationship) once the SeaFET has conditioned to the environment in which it is deployed. This, along with an explanation of why the Miller et al. results differ from Bresnahan et al., would be a useful analysis for the existing users of SeaFETs.

2) Characterize environmental variability: While the authors include a thorough explanation of how they minimized the impact of time/space sampling mismatch between the SeaFETs and the various independent validation measurements, it would be useful to develop an estimate of the environmental variability within these time/space constraints. This should be subtracted from (or considered somehow in) the sensor uncertainty estimates. An example of this type of assessment of how environmental variability impacts sensor field evaluations is summarized in the following and its associated ACT pCO2 sensor reports: Tamburri et al., 2011: Alliance for Coastal Technologies: Advancing Moored pCO2 Instruments in Coastal Waters. Marine Technology Society Journal, 45, 43-51.

Minor comments/edits:

Line 124: Elaborate; what does non-controlled source water conditions mean?

Line 437: State the impact of 0.21oC discrepancy on pH.

Lines 472-474: Improved accuracy for the unconditioned vs conditioned calibration based on what? The inter-sensor anomaly seems to be less in Fig 5 (conditioned) compared to Fig 4 (unconditioned), which is also shown in Fig 6.

Lines 680-683: While spectral analysis is a powerful tool for identifying drivers that are periodic or regular in nature, it will not characterize many phenomenon in the coastal environment such as storms or biological productivity/respiration. These types of events may impact the range over which a multi-point calibration should be made. This caveat should be included when suggesting spectral analysis as a tool for developing a multi-point calibration scheme in an environment with stochastic events.

Line 687: Define M2.

Figure 3: The temperature difference here could be misleading to the reader. It is important to be transparent by stating in the caption that the SeaFET was not fully submerged in the tank, making it susceptible to air temperature fluctuations unlike the BoL, which was measuring only tank water temperature.

---

## Referee Comment (RC2) · Anonymous Referee #2 · 5 Jun 2018

SeaFET instruments are now routinely used by the oceanographic community to measure pH in a variety of marine and coastal environments. In order to obtain useful data the instruments need to be calibrated and their performance assessed prior to, during and post deployment. This paper evaluates the performance of four such instruments under several different measurement environments and data work up procedures, and thereby provides an assessment of these particular measurements, and a procedure for other users to obtain the best data from a deployment.

This is a nice, tidily written paper, the figures and tables are relevant and easy to read and follow. As a SeaFET user myself, I found this to be a very useful manuscript,

with procedures that I can use to improve my own practice. However, there are no substantially new concepts presented.

I recommend that this manuscript be accepted with minor revisions, as detailed below.

Specific comments

Several methods of assessing data quality have been used – variability, accuracy (= integrated uncertainty), uncertainty, "true pHt", variance, RMSE, Standard deviation of duplicate samples, mean anomaly . . .Although Section 2.5.1 and 3.5 describes some of these terms in some detail, I found it difficult to assess the performance of the instruments and was distracted by the variety of terms. The sentence starting line 576 is a good example of this ". . . can provide and accurate measurement of pHt. . .. executed with high precision."

I suggest a table defining how and in what circumstance each term is used.

Line 257 specify austral winter

Lines 343, 400 why are the calibration coefficients on the header file and the CD-ROM different? If they are different how can the correct one be verified?

Line 410 SeaFET397 emerged from the tank for 24 hours. Did the pH sensor dry out? And if so, how was it reconditioned.

Line 467, the absolute difference of 2.83 oC is large in this context. How did you decide what temperature to use. Do you have a recommendation around calibration of the SeaFET temperature sensor?

Line 497 How was duration of the conditioning period determined, ie the width of the blue box in Figure 6. The 14 days indicated in Figure 6 is a long time

Line 512 the sentence starting "There was no clear distinction in greater accuracy.." does not make sense to me. Please rewrite this.

[Figure]

Line 632 The sentence starting "For instances of . . ." makes no sense, please reword.

Line 648 You state that ". . .the potential uncertainties calculated in this study represent the upper limit of an average uncertainty. . . .." How are you able to ascertain that this is an upper limit?

Line 654 You begin to discuss the effects of errors in the temperature measurement, but stop short of making any recommendations. This section should be tightened up, to go beyond a description of your own deployments.

Line 667 ". . .expanding the scope of pH variability. . ." this does not make sense

It would be useful to include a bullet pointed list of recommendations in the Conclusion

Was there any evidence of biofouling affecting the pH measurement during any of the deployments? Would you be able to determine the effect of this with your calibration strategy, and do you have any recommendations on how to identify this problem?

References – These are complete and up to date.

Figures – In general these are clear and helpful. I do not understand, however, the difference between Figure 4 and Figure 5. They are the same data sets, but Figure 4 is for "before they were conditioned", and "Figure 5 is for 'conditioned'". Does this refer to the way they were calibrated? Please clarify in the Figure caption.

---

## Author Comment (AC1) · 5 Jul 2018

Major comments: 1) Assess uncertainty if following today's existing SOP: While this study clearly shows the utility of single- and multi-point calibrations in these three coastal systems, it would be useful to discuss the uncertainty observed when following the existing SeaFET standard operating procedure (SOP). I believe the SOP described by Bresnahan et al. 2014 is to use the factory calibration, but correct the dataset to some independent measure of "true" pH (e.g., discrete bottle samples, pH derived from other biogeochemical sensors combined with locally-constrained carbon system algorithm or TA-S relationship) once the SeaFET has conditioned to the environment in

which it is deployed. This, along with an explanation of why the Miller et al. results differ from Bresnahan et al., would be a useful analysis for the existing users of SeaFETs.

Response

Specific to Bresnahan et al. (2014), a single-point in situ calibration was performed after the sensor was conditioned. This is different than the factory calibration performed by Sea-Bird. Factory calibration by Sea-Bird is conducted in a highly controlled tank setting void of natural conditions.

We agree with the reviewer that the current SOP described by Bresnahan et al. (2014) is slightly unclear as multiple cross-reference methods are proposed when determining SeaFETTM uncertainty (e.g., discrete bottle samples, pH estimated from O2 or PCO2). Thus, uncertainty will vary depending on validating source. We recognize the reviewer's comment regarding the utility of further comparing our uncertainty with that of Bresnahan et al. (2014), and have amended the manuscript to reflect this. See lines 677 - 687 in revised manuscript. Further, we refer the reviewer to lines 618- 636 which detail our assessment as to why some of our findings differ from what is currently in the primary literature.

We believe one utility of this manuscript is adding clarity to some of the ambiguity in terms and methods used in the primary literature regarding SeaFETTM operation and, thus, thank the reviewer for pointing out how we can improve by providing direct comparisons.

2) Characterize environmental variability: While the authors include a thorough explanation of how they minimized the impact of time/space sampling mismatch between the SeaFETs and the various independent validation measurements, it would be useful to develop an estimate of the environmental variability within these time/space constraints. This should be subtracted from (or considered somehow in) the sensor uncertainty estimates. An example of this type of assessment of how environmental variability impacts sensor field evaluations is summarized in the following and its associated

ACT pCO2 sensor reports: Tamburri et al., 2011: Alliance for Coastal Technologies: Advancing Moored pCO2 Instruments in Coastal Waters. Marine Technology Society Journal, 45, 43-51.

Response

We agree with the reviewer that providing these estimates is beneficial. We have amended the manuscript to incorporate how rapid changes in salinity and temperature can affect pH measurements. However, this was done based on our data, and we caution operators that our deployment sites are not representative of different regions.

In addition, we chose to forgo adding an additional figure or amending figure 10 because environmental variability is specific to an operator's location and not specifically an intrinsic uncertainty with the SeaFETTM. For this reason, we feel a detailed description of environmental variability in the text (see lines 638 -654) responding to the potential uncertainties that could arise from the effects of rapid environmental variability are sufficient.

Minor comments/edits: Line 124: Elaborate; what does non-controlled source water conditions mean?

Response

We have elaborated this line to be clearer. Non-controlled source water refers to non-manipulated seawater.

Line 437: State the impact of 0.21 C discrepancy on pH.

Response

This line has been amended to express the uncertainty from this temperature discrepancy.

Lines 472-474: Improved accuracy for the unconditioned vs conditioned calibration based on what? The inter-sensor anomaly seems to be less in Fig 5 (conditioned)

compared to Fig 4 (unconditioned), which is also shown in Fig 6.

Response

We agree with the reviewer that this may have been a bit unclear. We have amended the manuscript (see lines 481 - 482) to state the accuracy improved relative to the discrete reference samples. The reviewer is correct, the inter-sensor variability is less when calibrated and compared after the sensors were conditioned. When we discuss accuracy, we are specifically referring to sensor pH vs. discrete reference samples. We have provided a table based on reviewer # 2's comments to make clear the terminology used throughout the manuscript is consistent and clear.

Lines 680-683: While spectral analysis is a powerful tool for identifying drivers that are periodic or regular in nature, it will not characterize many phenomenon in the coastal environment such as storms or biological productivity/respiration. These types of events may impact the range over which a multi-point calibration should be made. This caveat should be included when suggesting spectral analysis as a tool for developing a multi-point calibration scheme in an environment with stochastic events.

Response

We agree with the reviewer that spectral analysis—while useful in some systems—may not be as insightful in indicating the main drivers of pH. However, if no observable pattern is distinguishable using spectral analysis, this in itself, will help indicate which calibration method is appropriate (e.g., in situ single-point or multi-point). We have amended the manuscript to reflect this comment according to the reviewer's concern. Please see lines 733 – 738 in the revised manuscript.

Line 687: Define M2.

Response

We have removed the M2 term and simply stated "mixed semi-diurnal."
Figure 3: The temperature difference here could be misleading to the reader. It is important to be transparent by stating in the caption that the SeaFET was not fully submerged in the tank, making it susceptible to air temperature fluctuations unlike the BoL, which was measuring only tank water temperature.

Response

We appreciate this comment, and have amended the figure caption to make clear that the top portion of the sensor may have been exposed to air temperature fluctuations.

---

## Author Comment (AC2) · 5 Jul 2018

Specific comments Several methods of assessing data quality have been used – variability, accuracy (= integrated uncertainty), uncertainty, "true pHt", variance, RMSE, Standard deviation of duplicate samples, mean anomaly . . .Although Section 2.5.1 and 3.5 describes some of these terms in some detail, I found it difficult to assess the performance of the instruments and was distracted by the variety of terms. The sentence starting line 576 is a good example of this ". . . can provide and accurate measurement of pHt. . .. executed with high precision."

I suggest a table defining how and in what circumstance each term is used.

[Figure]

Response

We thank the reviewer for the comment and have added a table defining some of the terminology used.

Line 257 specify austral winter

Response

We have amended this line to indicate winter in the northern hemisphere: boreal winter.

Lines 343, 400 why are the calibration coefficients on the header file and the CD-ROM different? If they are different how can the correct one be verified?

Response

After speaking with the engineers at Sea-Bird, I was instructed that the calibration coefficient on the CD-ROM was correct and the one written to the header file was not. After this conversation, I do not know their actions to resolve this issue, but they are aware of the problem. While I was instructed that the CD-ROM calibration coefficient is correct, we are hesitant to fully trust this response until they provide a more detailed response regarding this issue.

Line 410 SeaFET397 emerged from the tank for 24 hours. Did the pH sensor dry out? And if so, how was it reconditioned.

Response

It is not clear whether or not the electrodes dried out as the tank refilled before proper examination. Given the humidity in the room and in the tank, as well as a robust performance throughout the deployment, we do not believe the sensor was damaged. Since this failure occurred on April 8th and calibration did not occur until April 25th, any reconditioning required would have taken place within that 16 day window once the tank was refilled.

Line 467, the absolute difference of 2.83 C is large in this context. How did you decide what temperature to use. Do you have a recommendation around calibration of the SeaFET temperature sensor?

Response

We agree with the reviewer that this is a large temperature difference; however, we had more confidence in the Thermosalinograph readings given the history of reliability of this sensor in the community and at this specific location coupled to the BoL. In addition, there were multiple temperature probes monitoring incoming water throughout the hatchery that we cross-referenced. Suggestions as to accurately monitor temperature when calibrating the SeaFETTM are suggested in the manuscript. Please see lines 693 – 699.

Line 497 How was duration of the conditioning period determined, ie the width of the blue box in Figure 6. The 14 days indicated in Figure 6 is a long time

Response

Given that Bresnahan et al. (2014) indicate an approximate conditioning period of around 10 days, we were able to use this as a baseline to see when measurements stabilized over a several day period. This is how we were able to determine the conditioning period.

Line 512 the sentence starting "There was no clear distinction in greater accuracy.." does not make sense to me. Please rewrite this.

Response

This sentence has been rewritten for clarity based on the reviewer's comment.

Line 632 The sentence starting "For instances of . . ." makes no sense, please reword.

Response

We have rewritten the sentence to be clearer. Specifically, this sentence refers to the variance within discrete reference sample collection. That is, when collecting calibration samples, replicates will display a certain degree of variance for a "true pH" value, and this should be considered when calibrating the sensor. For example, we found a higher than desired discrepancy in our triplicate calibration samples taken for Kasitsna bay, so one replicate was thrown out, and duplicate calibration samples were used rather than triplicate for calculation of a "true pH."

Line 648 You state that ". . .the potential uncertainties calculated in this study represent the upper limit of an average uncertainty. . ..." How are you able to ascertain that this is an upper limit?

Response

We suggest that these are upper limits since our ranges of uncertainty fall within and, are, greater than what previous published results have found.

Line 654 You begin to discuss the effects of errors in the temperature measurement, but stop short of making any recommendations. This section should be tightened up, to go beyond a description of your own deployments.

Response

We appreciate the reviewer's concern, but believe that there is a clear recommendation regarding temperature. We state that it would be preferable to record temperature with a more robust instrument, or track temperature before deployment and apply an offset to the thermistor value.

Line 667 ". . .expanding the scope of pH variability. . ." this does not make sense

Response

This sentence has been rewritten based on the reviewer's comment.

It would be useful to include a bullet pointed list of recommendations in the Conclusion

[Figure]

Response

While we agree that this may be beneficial, we feel the main objective of this manuscript is to serve as an evaluation rather than a suggested best practices as this has already been done: Bresnahan et al. 2014 and Rivest et al. 2016.

Was there any evidence of biofouling affecting the pH measurement during any of the deployments? Would you be able to determine the effect of this with your calibration strategy, and do you have any recommendations on how to identify this problem?

Response

There was no evidence of biofouling affecting any of the SeaFETs. There was no evidence of biofouling at all for the Alaska SeaFETs, and the one at sentry shoal underwent maintenance during its deployment. In addition, this senor appeared to provide the most accurate measurements.

As far as identifying biofouling as interference, we do not offer suggestions, but this should be identifiable when you start to see a consistent drift in readings. This can be compared against other oceanographic data and the other electrode to verify biofouling, as well as a close physical inspection upon recovery. At the time biofouling is identified, calibration will need to be redone once the sensor is cleaned.

References – These are complete and up to date.

Response

No reply needed.

Figures – In general these are clear and helpful. I do not understand, however, the difference between Figure 4 and Figure 5. They are the same data sets, but Figure 4 is for "before they were conditioned", and "Figure 5 is for 'conditioned". Does this refer to the way they were calibrated? Please clarify in the Figure caption.

Response

The reviewer is correct, this does refer to how they were calibrated. We have amended the caption in figure 5 to make this clear.